# Conversion from coniferous to broadleaved trees can make European forests more climate-effective

Yi Yao [1] ✉, Petra Sieber [1], Mathias Hauser [1], Jonas Schwaab[2], Felix Jäger [1], Fulden Batibeniz [1,3,4], Meri Räty [5], Julia Pongratz[5,6], Martin Wild [1], Andrey Lessa Derci Augustynczik[7], Steven J. De Hertog [8], Verena C. Griess [9], Michael G. Windisch [1], Jun Ge [10], Alessio Collalti [11], Fulvio Di Fulvio[7], Petr Havlík[7] & Sonia I. Seneviratne [1]

The climate effectiveness of forestation in Europe is debated, as it may provide more warming via solar energy absorption than evaporative cooling. Since forests play an important role in European climate policy, it is necessary to explore potential solutions to this issue in a warmer world. Here, based on experiments conducted with a regional climate model under several forest change scenarios, we find that conversion from coniferous to broadleaved trees in currently forested areas can provide cooling for summer hot extremes (e.g., reducing the monthly mean daily maximum temperature in July over Continental Europe by 0.6 °C). The conversion can also mitigate the undesired warming impacts of forestation with present-day forest composition in most of Europe, e.g., reversing effects on the monthly mean daily maximum temperature in July over Continental Europe from +0.3 °C to −0.7 °C. This study highlights the importance of considering tree species in European forest policy development and suggests that the Northern and Central regions should be prioritised for forestation over the Western and Southern parts.

Forests cover around 40% of the land area of the European Union (EU)[1], and therefore the forest strategy plays an important role in the EU's climate policy, and its goal of net zero greenhouse gas emissions by 2050[2]. In addition to its biogeochemical (BGC) impacts (e.g., atmospheric carbon removal), changes in forests also have biogeophysical (BGP) impacts by altering the energy balance of the land surface through its influence on land surface properties[3,4]. Although the BGP impacts of historical anthropogenic land cover change are relatively small compared to the BGC impacts on the global scale, they can substantially affect and even dominate the local climate patterns in some regions[5–7]. Therefore, researchers suggest taking BGP impacts into account when developing forest strategies[8–11].

Previous efforts to study the BGP impacts of forest cover change generally agree on their meridional variations. The key feature is that forestation (afforestation and reforestation) decreases temperatures in tropical regions, and this cooling becomes less pronounced as the latitude increases, then switches to a warming in high-latitude regions (vice versa for deforestation)[4,9,12,13]. This is because although forests

[1]Institute for Atmospheric and Climate Science, Department of Environmental Systems Sciences, ETH Zurich, Zurich, Switzerland. [2]Institute for Environmental Planning, Leibniz Universität Hannover, 30419 Hanover, Germany. [3]Climate and Environmental Physics, Physics Institute, University of Bern, Bern, Switzerland. [4]Oeschger Centre for Climate Change Research (OCCR), University of Bern, Bern, Switzerland. [5]Ludwig-Maximilians-Universität München (LMU), Munich, Germany. [6]Max Planck Institute for Meteorology, Hamburg, Germany. [7]International Institute for Applied Systems Analysis, Laxenburg, Austria. [8]Q-ForestLab, Department of Environment, Ghent University, Ghent, Belgium. [9]Institute for Terrestrial Ecosystems, Department of Environmental Systems Sciences, ETH Zurich, Zurich, Switzerland. [10]School of Atmospheric Sciences, Nanjing University, Nanjing, China. [11]Forest Modelling Lab., Institute for Agriculture and Forestry Systems in the Mediterranean-National Research Council of Italy (CNR-ISAFOM), Perugia, PG, Italy. ✉e-mail: yi.yao@env.ethz.ch

generally absorb more solar radiation than grasslands or croplands[3,4,9,14–16], the impacts of forestation on upward energy fluxes (e.g., upwelling longwave radiation, sensible and latent heat fluxes) vary by latitudes[4,12,17–19]. More specifically, in low-latitude regions with ample water availability, forestation can substantially increase local evapotranspiration, converting a large amount of energy into latent heat. Conversely, in high-latitude regions, most of the additionally absorbed solar radiation becomes sensible heat and upwelling longwave radiation. Based on these results, several studies suggest that priority should be given to forestation in low-latitude regions to prevent local BGP warming from offsetting the BGC cooling[8,11].

Over Europe, forest cover increase leads to BGP warming in most areas[3,20], which reduces the attractiveness of forestation as a component of climate policy in this region. Observational studies have found that the transition from coniferous to broadleaved trees has cooling effects[9,12], suggesting that this warming can be prevented by forest species conversion. This may partly explain the latitudinal variations of forestation-induced BGP impacts, as coniferous forests dominate boreal and alpine regions, while tropical and temperate zones mainly consist of broadleaf forests[21]. Climatic suitability is an important reason for this distribution. However, Europe's profit-driven forestry policies have also contributed substantially to the composition of the continent's forests. During the last three centuries, foresters favoured high-profit tree species such as Scots pine and Norway spruce, reducing the fraction of broadleaf forests substantially, contributing to substantial warming, especially over Central Europe[22]. Another study extended to future periods confirms that there is a huge cooling potential by converting coniferous to broadleaf forests due to albedo increase[23].

Although existing studies provide valuable information on climate-compatible forestation, they are not without limitations. First, the algorithms used in observation-based studies[9,12] only allow quantifying local effects caused by direct forest changes, failing to explore the implications of large-scale forest cover and composition change and ignoring the remote impacts induced by atmospheric feedback (which can be dominant over some regions[19,24]). Second, in modelling studies[22,23], there is a lack of investigation into the impacts of the combination of forestation and forest species conversion, and the comparison of their individual contributions, especially under future warming scenarios. This study seeks to fill these gaps and answer two key questions: (1) Can species conversion in existing forests generate BGP impacts comparable to those of deforestation over Europe? If confirmed, this could enable similar BGP cooling effects while maintaining the BGC benefits. (2) Can combining forestation and species conversion prevent undesired afforestation-induced BGP warming in high-latitude regions? If feasible, this would offer a pathway for most regions to contribute to both local cooling through BGP effects and global cooling through BGC effects. In addition, near-surface air temperature was commonly used in previous research, but some studies revealed that forest changes-induced impacts on different temperature variables (like surface skin temperature and 2-meter air temperature) may vary, emphasising the need to consider the entire near-surface temperature profile[25,26].

To address these questions, we design a set of European forest cover change and forest species conversion experiments (Table 1), representing multiple forest management activities. For each experiment, we perform a simulation under a Shared Socioeconomic Pathway scenario (SSP3-7.0, as it represents a scenario at the high end of warming that current climate policy could lead to[27]) with a regional climate model (COSMO −CLM²) covering the period from 2015 to 2059. The impacts of those experiments on near-surface temperature, especially hot extremes, are systematically analysed to explore which strategies for European forests could be beneficial in terms of BGP effects. We also examine the impacts of forest changes on the temperatures of the land surface and the lowest atmosphere model level,

and analyse surface energy fluxes to identify the drivers of the changes in temperatures.

## Results

### Coniferous tree species dominate present-day European forests

The present-day land-use distribution (control: Ctl, from the Land-Use Harmonisation 2, LUH2 dataset[28]) shows that European forests are mainly located in high-latitude and high-altitude regions (Fig. 1a). Conversely, grasslands dominate the vegetation at low-latitude and low-altitude regions (Fig. 1b). In total, forests cover more than 29.1% of the land area. We select five sub-regions based on climate and environmental similarity[29] (Fig. 1e) for further analysis. Among forests, coniferous forests account for 72.9% (Alpine), 85.7% (Northern), 58.6% (Atlantic), 48.4% (Continental), 65.3% (Southern), and 69.9% in total over Europe (Fig. 1f).

### Forest species conversion mitigates summer hot extremes more substantially than deforestation

Converting all existing coniferous forests to broadleaf (Brd) forests can substantially reduce hot extremes (defined as the average daily maximum 2-meter temperature ($T_{air}$) across summer: Fig. 2f), by ≥0.5 °C over a large part of Europe. Compared to the cooling impacts caused by deforestation (Def: Fig. 2g), Brd induces a slightly more pronounced cooling (ranging from 0.1 to 0.5 °C) over most of mid- and high-latitude regions (Fig. 2h). However, in the Mediterranean region, compared to Def, Brd has less cooling effect, whose effect can exceed 0.5 °C in some grid cells (Fig. 2e). The cooling effects observed in both scenarios are strongly associated with reductions in summer mean net shortwave radiation ($SW_{net}$; Fig. S1j, k), primarily driven by increases in summer mean upwelling shortwave radiation ($SW_{up}$; Fig. S1b, c). Additionally, Brd causes a slight decrease in summer mean downwelling shortwave radiation ($SW_{down}$; Fig. S1f), whereas Def leads to an increase in many regions, potentially due to changes in cloud cover (Fig. S1g).

Although both Brd and Def reduce the summer mean $SW_{net}$, changes in summer mean daily maximum surface skin temperature ($T_{sfc}$, directly determining upwelling long-wave radiation, $LW_{up}$) under the two scenarios exhibit opposite patterns, with Brd leading to cooling and Def resulting in warming across most of the study area (Fig. 2b, c), possibly due to differences in the partitioning of land surface energy fluxes. Both Brd and Def reduce the summer mean sensible heat flux from the land to the atmosphere ($H_{up}$; Fig. S2j, k) as a result of decreased summer mean $SW_{net}$. However, Brd enhances the summer mean latent heat flux ($LE_{up}$) due to the higher evapotranspiration rate of broadleaf trees (Fig. S2b), whereas Def leads to a reduction in summer mean $LE_{up}$ across most of Europe (Fig. S2c). As a result, more energy at the land surface is released as summer mean $LW_{up}$ under Def (Fig. S2s), and at the same time, a decrease in summer mean $H_{up}$ (Fig. S2k) indicates a lower ability of the land surface to heat up the air, which leads to a decrease in summer mean daily maximum temperature of the lowest atmosphere level ($T_{atm}$). Different patterns emerge when considering daily mean and daily minimum temperatures (Figs. S3, S4), such as in Southern Europe, both summer mean daily minimum $T_{air}$ and $T_{atm}$ increase under Def (Fig. S4g, k). This may be explained by an increase in nighttime $H_{up}$ resulting from reduced $LE_{up}$ under Def, which becomes the dominant factor in the absence of shortwave radiation.

To better quantify and understand the mechanisms underlying the impacts of forest changes on temperature, we calculate the summer mean energy fluxes and temperatures over five sub-regions (Fig. 3a–c). For example, in the Atlantic region, both Brd and Def increase summer mean $SW_{up}$ from 25.82 to 28.40 and 29.99 W m$^{-2}$, respectively, due to the albedo increase associated with the conversion of coniferous forests to broadleaf forests or grassland. In contrast, Brd slightly decreases summer mean $SW_{down}$ by 185.58 to 184.71 W m$^{-2}$,

**Table 1 | Experimental design**

| Experiments | Description | Objectives |
|---|---|---|
| Ctl | Present-day natural vegetation distribution | This experiment is used for reference to calculate the impacts of forest changes. Its sub-grid cell level outputs (e.g., forest and grassland) are also used for evaluation (see Supplementary Note 1). |
| Aff | Replacing all grasslands with forests, without changing the fraction of tree types. If there is no forest in the present-day land use, the average fraction at the same latitude is applied. | This experiment is used to detect the impacts of forestation (afforestation and reforestation) based on local present-day forest composition. It is also used for evaluation together with Def (see Supplementary Note 1). Its outputs are analysed in the main text. |
| Def | Replacing all forests with grasslands, without changing the fraction of grass types. If there is no grassland in the present-day land use, the average fraction at the same latitude is applied. | This experiment is used to detect the impacts of deforestation based on local present-day grassland composition. It is also used for evaluation together with Aff (see Supplementary Note 1). Its outputs are analysed in the main text. |
| Brd | Switching all coniferous forests to broadleaf forests, without changing the relative fraction of the five broadleaf species. If there is no broadleaf forest in the present-day land use, the average fraction at the same latitude is applied. | This experiment is used to detect the impacts of the transition from coniferous to broadleaf trees in the present-day forest. Its outputs are analysed in the main text. |
| Ndl | Switching all broadleaf forests to coniferous forests, without changing the relative fraction of the three coniferous species. If there is no coniferous forest in the present-day land use, the average fraction at the same latitude is applied. | This experiment is mainly used for the comparison with Brd. Its outputs are not analysed in the main text. |
| AfB | Switching all grasslands and coniferous forests to broadleaf forests, without changing the relative fraction of the five broadleaf species. If there is no broadleaf forest in present-day land-use, the average fraction at the same latitude is applied. | This experiment is used to detect the impacts of the combined changes of forestation and the conversion from coniferous to broadleaf trees. It is also used for evaluation together with AfN (see Supplementary Note 1). |
| AfN | Switching all grasslands and broadleaf forests to coniferous forests, without changing the relative fraction of the three coniferous species. If there is no coniferous forest in present-day land-use, the average fraction at the same latitude is applied. | Mainly used for the comparison with AfB. It is also used for evaluation together with AfB (see Supplementary Note 1). Its outputs are not analysed in the main text. |

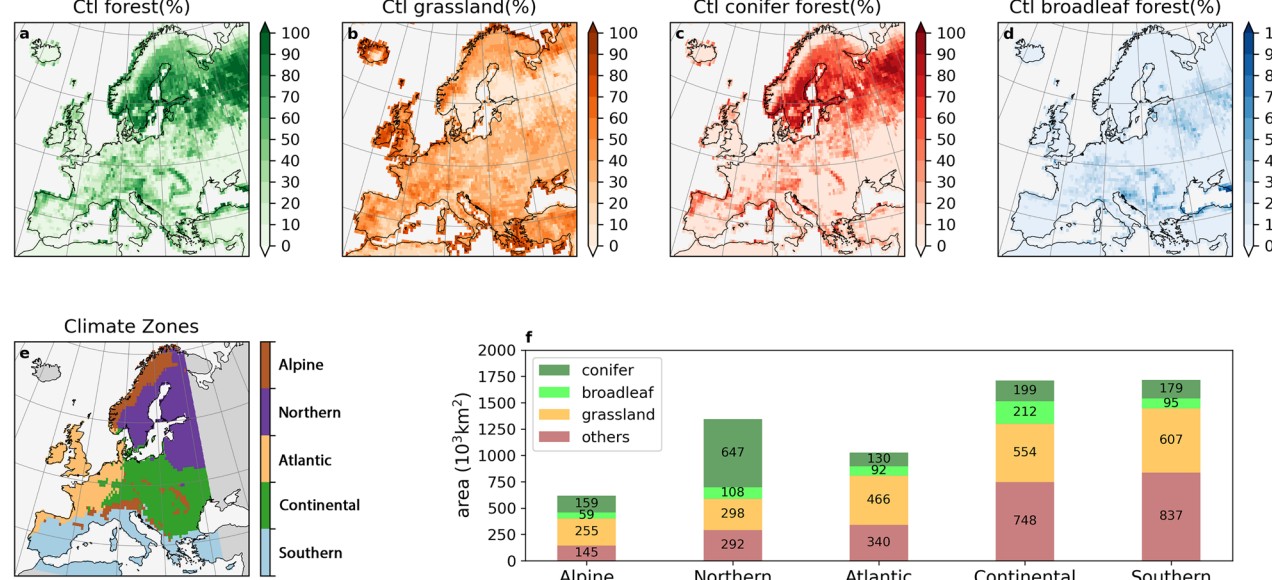

**Fig. 1 | Distribution of forests, grasslands, coniferous and broadleaf forests and land-use changes in idealised scenarios.** Present-day distribution (fraction of grid cell area) of forests (**a**), grasslands (**b**), coniferous forests (**c**), and broadleaf forests (**d**), used for the control (Ctl) simulation. Grid cells corresponding to five climate regions (Alpine, Northern, Atlantic, Continental and Southern) used for time series analysis (**e**). Present-day total areas of coniferous forests, broadleaf forests, grasslands, and other land use types in the five zones (**f**).

whereas Def increases it from 185.58 to 187.32 W m$^{-2}$, likely owing to changes in cloud cover. Consequently, Brd results in a slightly lower summer mean SW$_{net}$ than Def in summer. Both scenarios have very small effects on summer mean LW$_{up}$ and ground flux (G$_{down}$), so the increase in summer mean SW$_{up}$ is redistributed among LE$_{up}$, H$_{up}$, and LW$_{up}$. Under Brd, summer mean LE$_{up}$ increases substantially as broadleaf trees generally consume more water, leading to decreases in both H$_{up}$ and LW$_{up}$, thereby producing cooling effects on all three temperatures. In contrast, under Def, both summer mean LE$_{up}$ and H$_{up}$

decrease, and LW$_{up}$ is therefore enhanced. As a result, summer mean daily maximum T$_{sfc}$ increases, whereas the other two temperatures decrease. Slight differences in these temperature responses may occur when examining other temperature metrics, which can be attributed to the diurnal variability in energy fluxes.

Averaged within five sub-regions, monthly mean daily maximum temperature (Fig. 4), daily mean temperature (Fig. S5), and daily minimum temperature (Fig. S6) are most substantially affected in the Northern region, followed by the Alpine region. In the Northern region

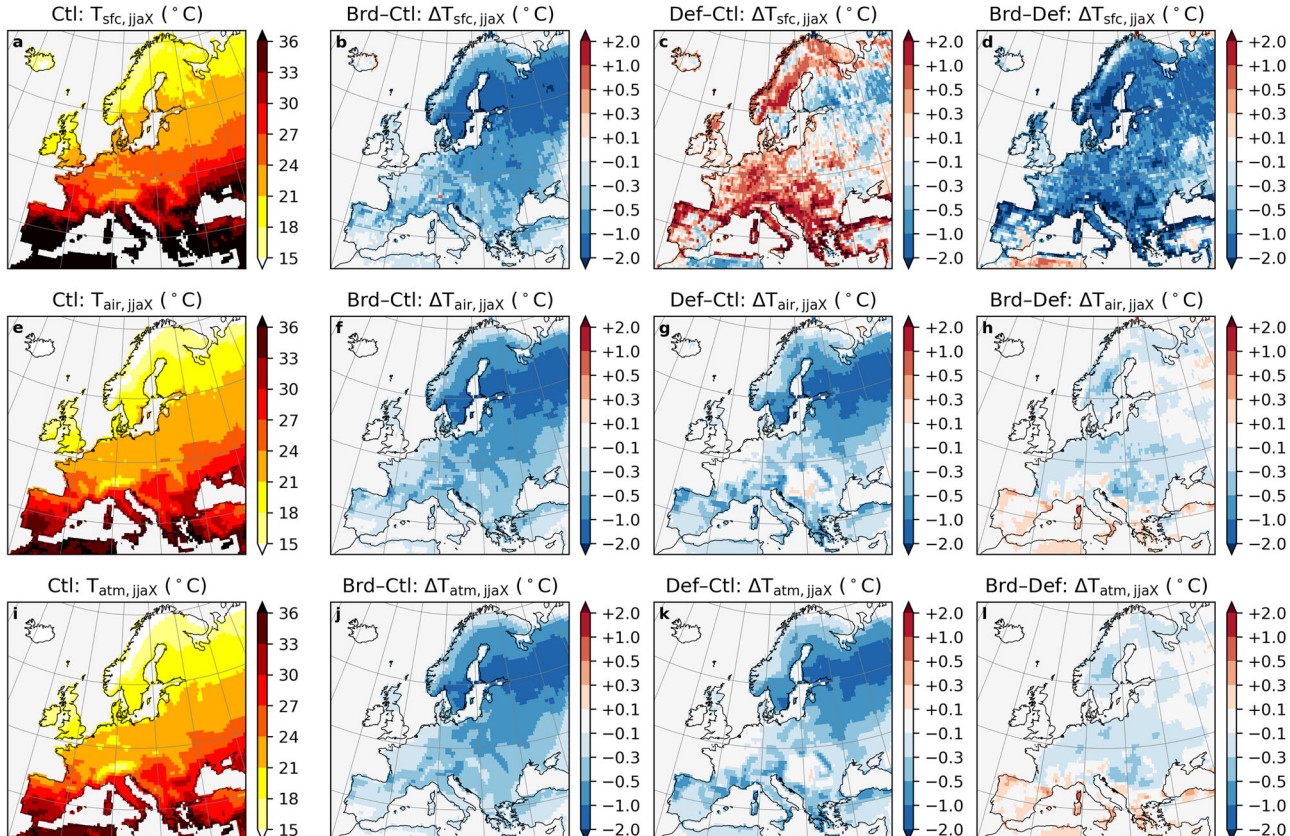

**Fig. 2 | Conversion from coniferous to broadleaf trees in existing forests can mitigate summer hot extremes more than deforestation.** Multi-year (2025–2059) summer (June, July, and August) mean daily maximum land surface temperature ($T_{sfc,jjaX}$: **a**), 2-meter air temperature ($T_{air,jjaX}$: **e**), and the temperature at the lowest atmospheric level ($T_{atm,jjaX}$: **i**), under the present-day forest scenario (Ctl). Changes in these temperatures, respectively, under the conversion from coniferous to broadleaf forests scenario (Brd-Ctl: **b**, **f**, **j**), under the deforestation scenario (Def-Ctl: **c**, **g**, **k**), and the difference between the two scenarios (Brd-Def: **d**, **h**, **l**).

(Fig. 4c, d, S5c, d, and S6c, d), Def results in reductions of monthly mean daily maximum, mean, and minimum $T_{air}$ by approximately 2.8 °C, 1.9 °C, and 1.3 °C, respectively, in April and May. This substantial cooling may be related to the snow-radiation feedback, which may decrease in a warming world. However, these cooling effects diminish and nearly disappear during the summer. In contrast, Brd provides a relatively consistent cooling effect of about 1.0 °C, 0.5 °C, and 0.2 °C from March to July for monthly mean daily maximum, mean, and minimum $T_{air}$, respectively. A similar temporal pattern is observed in other regions, albeit with smaller magnitudes. For instance, Brd reduces monthly mean daily maximum $T_{air}$ by ~0.6 °C in July in the Continental region, while Def only has a cooling of around 0.4 °C (Fig. 4g, h). These findings suggest that changing the management of present-day forests (Brd) may offer an effective strategy for mitigating summer heat stress, although the potential for cooling is limited in the Atlantic and Southern regions.

When examining changes in the monthly mean daily maximum, mean, and minimum $T_{sfc}$, $T_{air}$, and $T_{atm}$, distinct seasonal patterns emerge under Def (Fig. 4, S5 and S6). In general, changes in monthly mean daily maximum $T_{air}$ and $T_{atm}$ are closely aligned, whereas $T_{sfc}$ exhibits markedly different behaviour. For instance, over the Atlantic region, monthly mean daily maximum $T_{sfc}$ increases by approximately 0.75 °C in August, while both monthly mean daily maximum $T_{air}$ and $T_{atm}$ show slight decreases (Fig. 4f). In contrast, under Brd, the seasonal patterns of monthly mean daily maximum $T_{sfc}$ remain broadly consistent with those of monthly mean daily maximum $T_{air}$ and $T_{atm}$, despite some differences in magnitude. Similar patterns are also observed for the mean and minimum temperatures (Figs. S5 and S6). As discussed earlier, these differences are

closely linked to the partitioning of surface energy fluxes, with the majority of the energy out-flux being released as upwelling longwave radiation ($LW_{up}$; Fig. S7). More specifically, between April and October, Brd facilitates a greater magnitude of the latent heat flux ($LE_{up}$), whereas this is reduced in Def, inducing higher $LW_{up}$.

## Forestation with broadleaf trees can prevent local warming effects

Given the urgent need for net negative emissions to meet climate targets, forestation is considered an important strategy for atmospheric carbon dioxide removal[30,31], despite its possible BGP warming impacts. It is therefore crucial to explore the potential for mitigating undesired BGP impacts of forestation through forest management strategies. We design two forestation scenarios to investigate this further. In the first, we maintain the current coniferous and broadleaf composition (Aff). This forestation experiment increases summer mean daily maximum $T_{air}$ over the entire continent, particularly in the Mediterranean region, where warming exceeds 1.0 °C in most grid cells (Fig. 5d). This warming effect is primarily driven by the increase in summer mean $SW_{net}$ (Fig. S8i), which in turn results mainly from the decrease in $SW_{up}$ (Fig. S8a). Considering that this region is also the region with the highest summer mean daily maximum $T_{air}$ (Fig. 5e), Aff may substantially exacerbate local heat stress. Similar to Def, Aff has opposite impacts on summer mean daily maximum $T_{air}$ and $T_{sfc}$, with the cooling impact on $T_{sfc}$ exceeding 1.0 or even 2.0 °C in some grid cells (Fig. 5a, d). More specifically, compared to grassland, forests can absorb more shortwave radiation, but can also increase both turbulent fluxes (summer mean $LE_{up}$ and $H_{up}$), leading to a decrease in $LW_{up}$

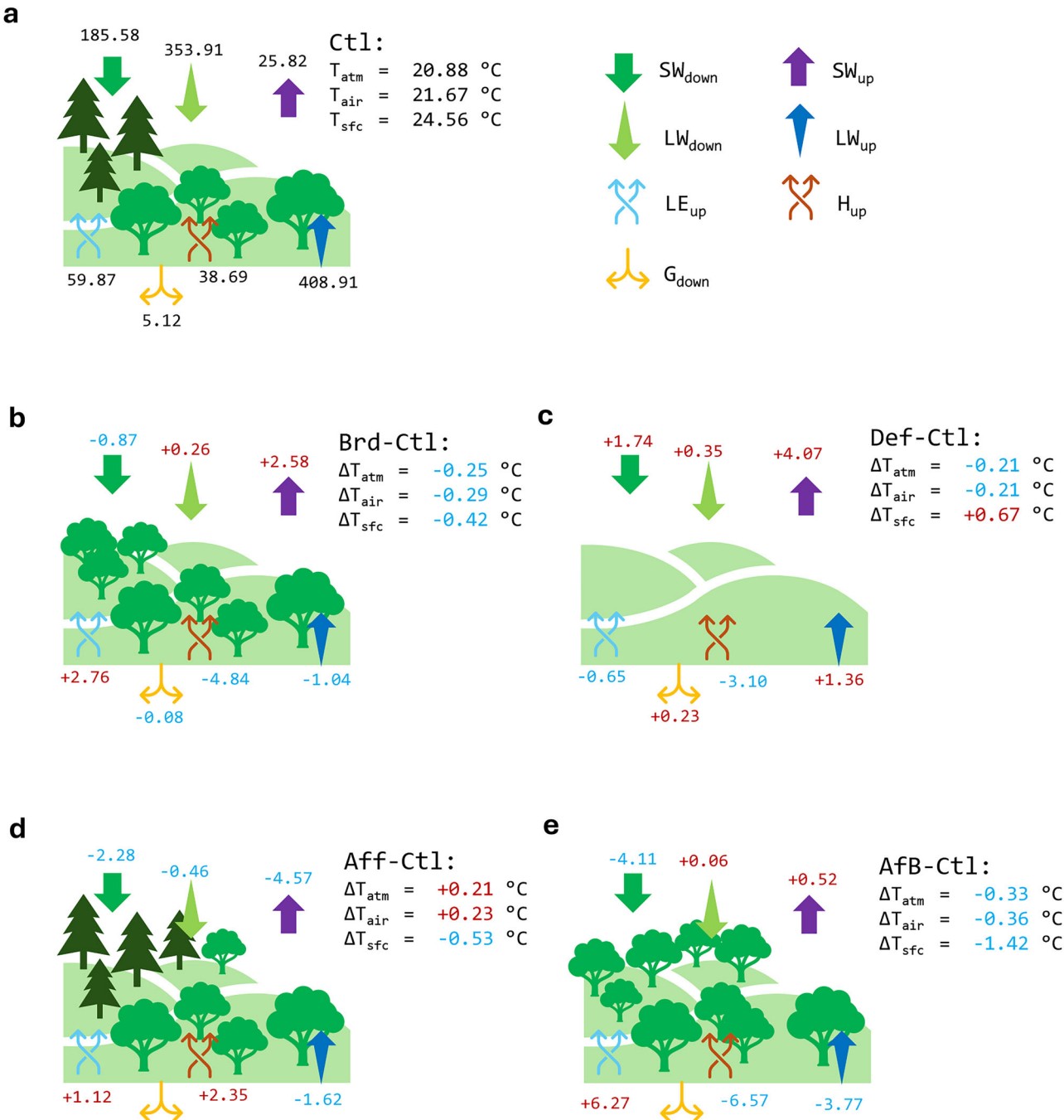

**Fig. 3 | Biogeophysical impacts of forest change scenarios driven by changes in energy fluxes.** Multi-year (2025–2059) summer (June, July, and August) mean daily maximum temperatures (of land surface: $T_{sfc}$; of 2-meter air: $T_{air}$; and the lowest atmosphere level: $T_{atm}$), and energy fluxes (down-welling shortwave radiation: $SW_{down}$; up-welling shortwave radiation: $SW_{up}$; down-welling longwave radiation: $LW_{down}$; up-welling longwave radiation: $LW_{up}$; latent heat flux from the land to the atmosphere: $LE_{up}$; sensible heat flux from the land to the atmpsphere: $H_{up}$; and ground flux from the land surface to the ground: $G_{down}$) averaged over the Atlantic region (see Fig. 1e) under the present-day forest scenario (Ctl: **a**), and the difference compared to Ctl under the conversion from coniferous to broadleaf forests scenario (Brd: **b**), under the deforestation scenario (Def: **c**), under the forestation scenario (Aff: **d**), and under the combining forestation and conversion from coniferous to broadleaf forests scenario (AfB: **e**). Numbers in blue indicate there is a decrease compared to the corresponding numbers under Ctl, and vice versa for numbers in red. Other temperatures (daily mean and minimum temperatures) and the results of other regions can be found in Table S1–5. Some icons used in this figure are from Microsoft PowerPoint, used under license.

(Fig. S9a, i). However, increased $H_{up}$ substantially heats up the air. Contrarily, Aff has a slight cooling impact on the multi-year summer mean daily mean temperature for all three temperature variables in the Southern part of the study area (Fig. S8), and this cooling expands to most of Europe in terms of the multi-year summer mean daily minimum temperature (Fig. S9).

In addition to Aff, we devise a second scenario in which all coniferous forests are converted to broadleaf after forestation (AfB). This scenario can reduce summer mean $SW_{net}$ over most of Europe, primarily due to a decrease in $SW_{down}$, possibly associated with enhanced cloud cover (Fig. S10j, f). As a result, AfB can help to avoid this forestation-related warming in most regions, resulting in cooling

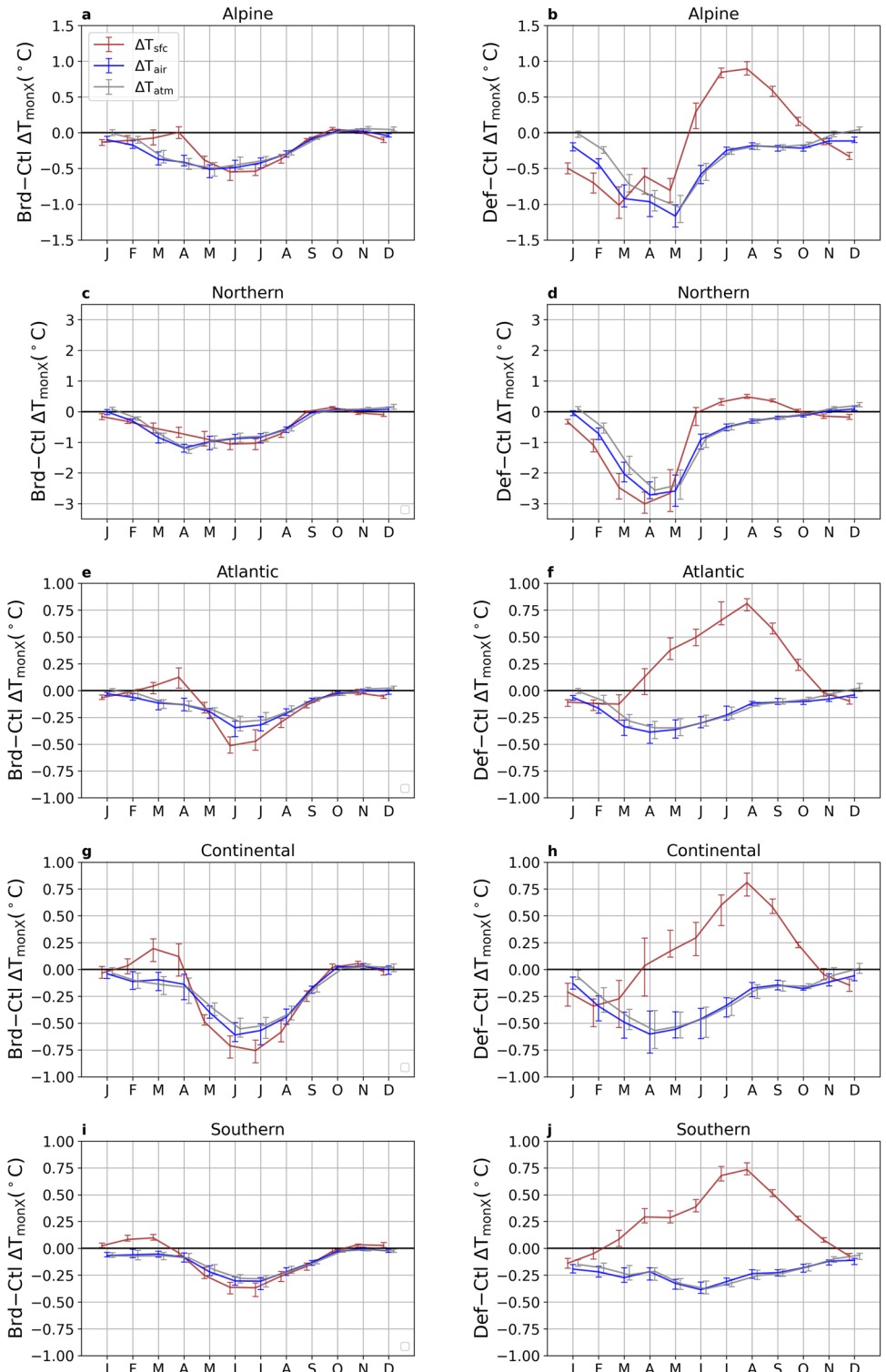

**Fig. 4 | Different seasonal patterns of biogeophysical impacts of forest composition change and deforestation across five regions.** Changes in multi-year mean (2025-2059) monthly mean daily maximum temperature ($T_{monx}$) induced by the conversion from coniferous to broadleaf forests scenario (Brd-Ctl) (**left column**) and the deforestation scenario (Def-Ctl) (**right column**). The values shown are the regionally averaged temperature change (minus the outputs from the Ctl simulation) over five regions: Alpine (**a, b**), Northern (**c, d**), Atlantic (**e, f**), Continental (**g, h**), and Southern (**i, j**) (see Fig. 1e). The ranges indicate the 25th and the 75th percentiles of $T_{monx}$ during the period 2025-2059.

effects on summer mean daily maximum $T_{air}$ across the majority of Europe, with reductions ranging from -0.5 to -2.0 °C (Fig. 5e). However, in the Mediterranean region, a warming impact persists (Fig. 5e), which may be related to the fact that the dominant present-day natural vegetation in this region is grassland, and the change from grassland to broadleaf forest leads to more $SW_{net}$. By comparing the AfB and Def experiments (Fig. 5h), we find that converting grassland to broadleaf forest induces warming in this region. The increase in summer mean $SW_{net}$ (Fig. S10l) cannot be fully offset by the rise in $LE_{up}$, as occurs in other regions, resulting in an accompanying increase in $H_{up}$ (Fig. S11d,

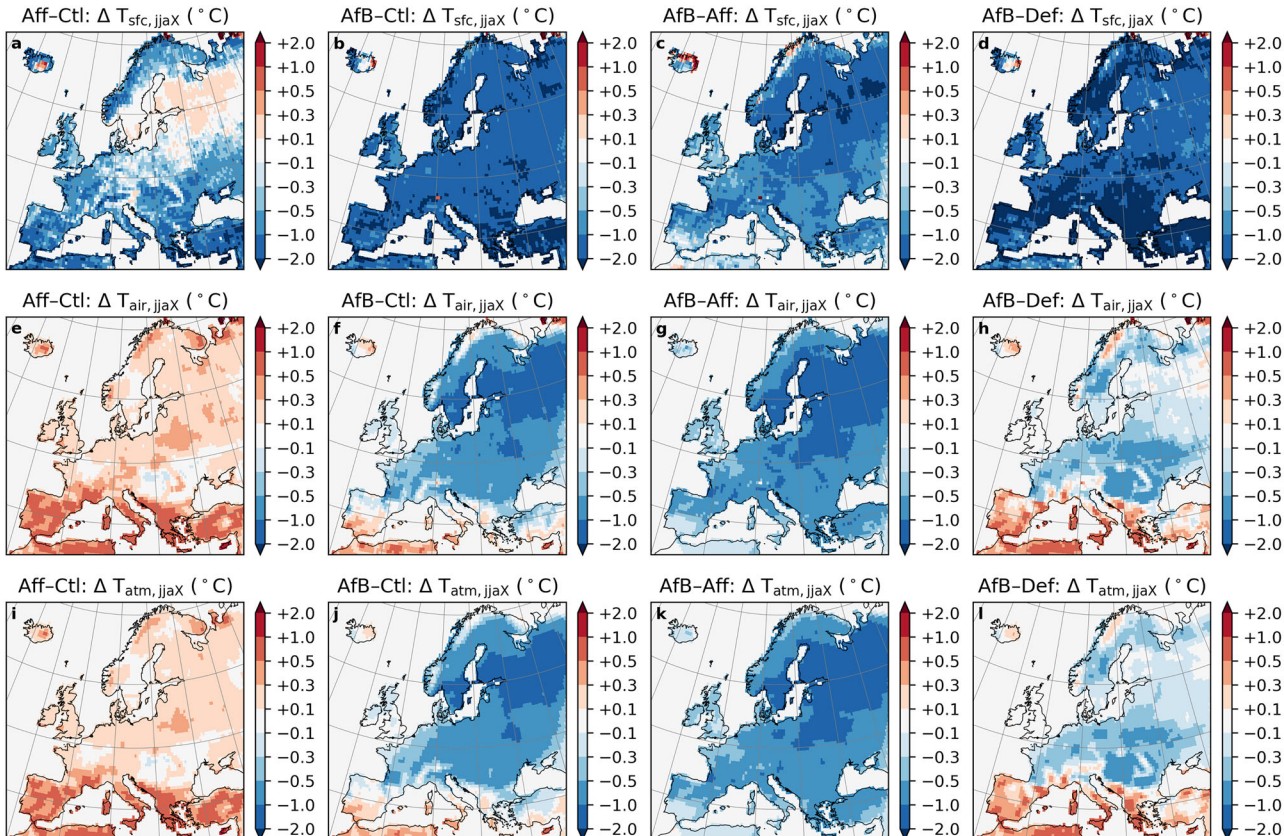

**Fig. 5 | Forest composition conversion can mitigate forestation-induced warming.** Changes in multi-year (2025-2059, compared to the experiment Ctl) summer (June, July, and August) mean daily maximum land surface temperature ($T_{sfc,jjaX}$: **a–d**), 2-meter air temperature ($T_{air,jjaX}$: **e–h**), and the temperature at the lowest atmospheric level ($T_{atm,jjaX}$: **i–l**) under the forestation scenario (Aff-Ctl: **a, e, i**), by the combining forestation and conversion from coniferous to broadleaf forests scenario (AfB-Ctl: **b, f, j**), the difference between the two scenarios (AfB-Aff: **c, g, k**), and the difference between AfB and the deforestation scenario (AfB-Def: **d, h, l**).

**l**). The cooling impacts of AfB also exist on the multi-year summer mean daily mean and minimum temperatures, but with a smaller magnitude (Figs. S8–S9).

Across the Atlantic region, Aff reduces both summer mean $SW_{down}$ (185.58 to 183.30 W m$^{-2}$, possibly due to increased cloud cover) and summer mean $SW_{up}$ (25.82 to 21.25 W m$^{-2}$, likely resulting from an increase in albedo), thereby decreasing $SW_{net}$ (Fig. 3a, d). Concurrently, summer mean $LE_{up}$ and $H_{up}$ both increase, contributing to a reduction in $LW_{up}$. This combination lowers summer mean daily maximum $T_{sfc}$ while increasing $T_{atm}$ and $T_{air}$. In contrast, AfB markedly decreases summer mean $SW_{down}$ (185.58 to 181.47 W m$^{-2}$, likely due to a substantial increase in cloud cover associated with enhanced evapotranspiration) and slightly increases $SW_{up}$ (25.82 to 26.34 W m$^{-2}$), producing a pronounced decline in $SW_{net}$ (Fig. 3a, e). The substantial increase in summer mean $LE_{up}$, reflecting the higher water consumption of broadleaf forests compared to coniferous forests and grassland, leads to decreases in both $H_{up}$ and $LW_{up}$, and consequently lowers all three temperature metrics. Results for other temperature variables and regions can be found in Supplementary Tables S1–S5.

Results for monthly mean daily maximum, mean, and minimum $T_{air}$ (Fig. 6, S12 and S13) indicate that AfB generally leads to more cooling than Aff during the summer months. For example, in the Northern region, the cooling of the monthly mean daily maximum $T_{air}$ in July reaches approximately 1.0 °C for AfB, while Aff has almost no impact (Fig. 6c, d). Moreover, Aff leads to substantial spring warming of monthly mean daily maximum $T_{sfc}$ in Alpine and Northern regions, which can be partially offset by AfB due to a decrease in monthly mean $SW_{net}$ (Fig. S14a–d). Compared to Brd, AfB provides a more substantial maximum summer cooling effect in all regions, such as an approximately 0.8 °C reduction for AfB versus 0.5 °C for Brd in the Continental region (Fig. 6g, h). An exception is observed in the Southern region for monthly mean daily maximum $T_{air}$, where AfB induces a warming effect in most months, and its cooling potential in summer is lower than both Def and Brd (Fig. 6j). This highlights the importance of carefully designing forestation strategies in this region. In general, considering the opposing temperature effects of forestation with broadleaf and needleleaf trees is essential to avoid warming effects. Specifically, forestation with broadleaf trees could be a solution that provides both BGP and BGC cooling benefits across a wider range of regions, rather than only tropical regions.

## Discussion

### Suitability of the model COSMO-CLM² and designed scenarios

We employed a new version of a regional climate model with state-of-the-art implementation of natural vegetation (COSMO-CLM²) to explore climate-effective forest strategies in a warming future (SSP3-7.0). By comparing the results of forestation and deforestation to the experiments under the framework of the Land Use and Climate Across Scales (LUCAS) regional climate model intercomparison[3], we find that in summer the COSMO-CLM aligns closely with the multi-model mean results. Another modelling-based study[32] also confirmed that converting coniferous to deciduous forests can effectively reduce the intensity of heat extremes over several grid cells in Europe, but this effect was minor in Scandinavian regions. This spatial discrepancy aligns with an observationally-based study[33] but differs from the results of the present work. The discrepancy may stem from differences in study periods, as the background climate in this study is warmer, potentially amplifying the evaporative cooling associated with forest-type changes. Given the

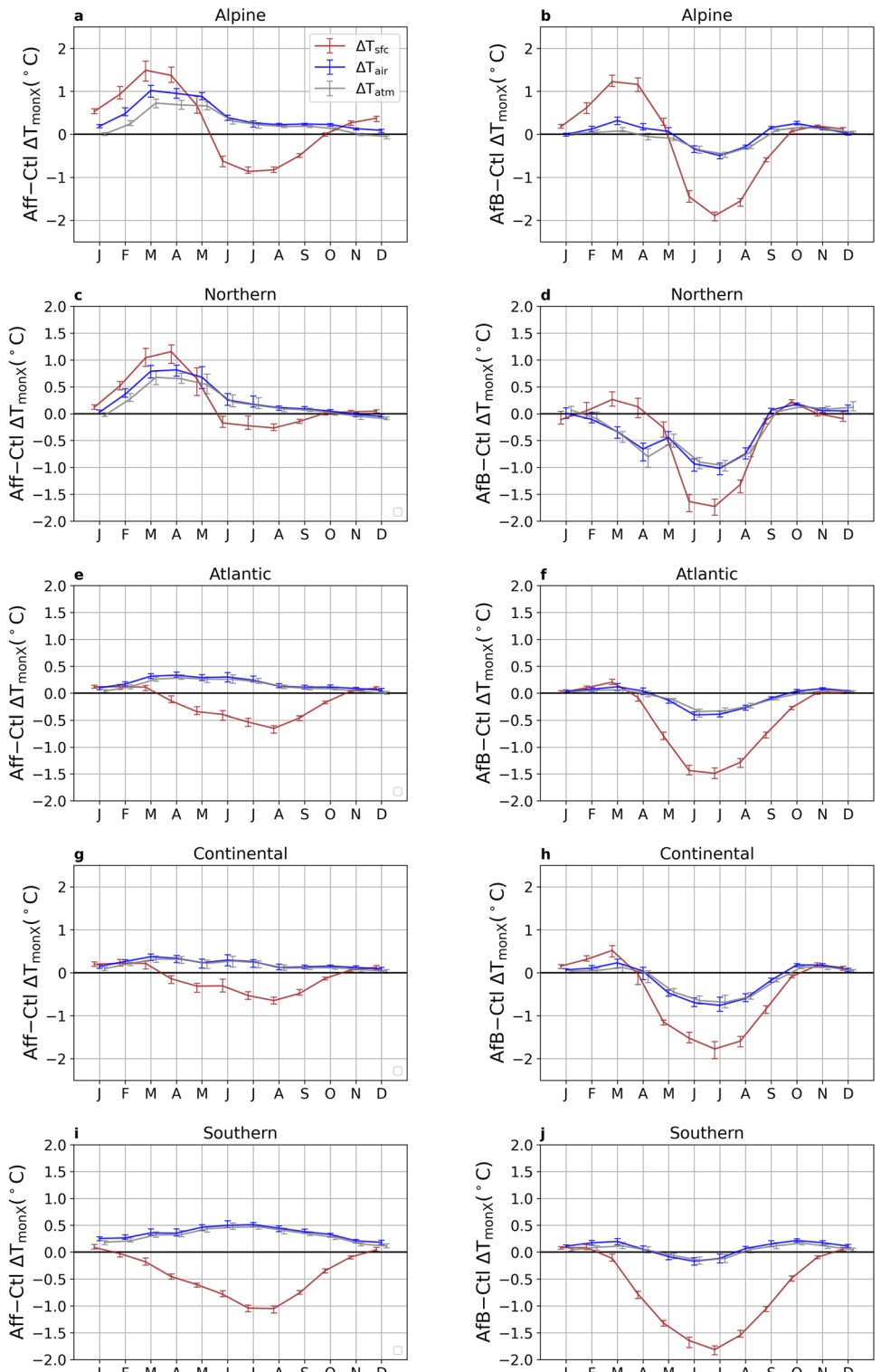

**Fig. 6 | Spatially variant seasonal pattern of forestation-induced biogeophysical impacts across five regions.** Changes in multi-year mean (2025-2059) monthly mean daily maximum temperature ($T_{monx}$) induced by the forestation scenario (Aff-Ctl) (**left column**) and the combining scenario of forestation and the conversion from coniferous to broadleaf forests (AfB-Ctl) (**right column**). The values shown are the regionally averaged temperature change (minus the outputs from the Ctl simulation) over five regions: Alpine (**a, b**), Northern (**c, d**), Atlantic (**e, f**), Continental (**g, h**), and Southern (**i, j**) (see Fig. 1e). The ranges indicate the 25th and the 75th percentiles of $T_{monx}$ during the period 2025-2059.

various forest change scenarios and subgrid-cell-level outputs, this study also facilitates comparison with previous observation-based findings. An evaluation of COSMO-CLM²'s performance in simulating the BGP impacts of forest changes is conducted based on multiple observation-based datasets[9,12,34] (see Supplementary Note 1). The comparison reveals that the model can reproduce the sign and magnitude of BGP impacts induced by forestation, deforestation, and forest species change at a satisfying level, especially for summer hot extremes.

Forest change scenarios designed for this study include forestation, deforestation, the conversion between coniferous and broadleaf forests, and their combinations. Due to the limited representation of forestry in CLM5, the impacts of many other common forest management activities, such as harvesting and forest health improvements, are not comprehensively explored. In this study, as we only focus on the BGP impacts, the deforestation experiment can, to some extent, approximate the effects of harvesting, and a group of sensitivity experiments (see Methods and supplementary Note 2) can approximate forest health change. Thus, we believe that this study provides valuable insights for the development of forest strategies in Europe.

### History and future of European forestry policy

The land-use dataset used in this study (LUH2) reveals that Europe has conifer-dominated forests, and these numbers are consistent with a recent report on the state of forests in Europe[35]. This report further highlights that two species, namely pine and spruce, together accounted for more than half of Europe's growing stock of timber in 2020. This is a result of tree species changes in the last few centuries, as indicated by a multi-source reconstruction of European forest management from 1600 to 2010[36]. It shows that the forest area stopped shrinking around the year 1850, and has recovered to a similar level as in the year 1600. However, the unmanaged forest area kept decreasing, and it only accounts for a very small fraction of the current European forests. This switch from unmanaged to managed forest also caused drastic changes in tree species composition, with a predominant shift from broadleaf to coniferous forests across an area exceeding 400.000 km$^2$. Consequently, the fraction of coniferous forests changed from less than a third in 1850 to more than half in 2010. Thus, the dominance of coniferous trees in present-day forests can be largely attributed to human influence, driven by the increasing demand for timber and other wood products[35,37].

Results of this study suggest that, from a BGP cooling perspective, coniferous tree species should be de-prioritised in forest management strategies. Currently, forestry plays an important role in the European economy[38], so policies may be needed to incentivise foresters to shift from commercially valuable species to climate-friendly alternatives. Considering that European forest coverage is expected to expand to achieve net-zero emissions, it is also important to carefully select the regions for afforestation and reforestation. Our simulations show that forestation with broadleaf species in the Northern part of Europe can provide the biggest potential for hot extreme mitigation, followed by Central and Eastern Europe. However, the cooling benefits are limited or even reversed to warming effects in Western and Southern regions, suggesting that forestation may not be prioritised there.

### Additional considerations in forest policy

Two notable limitations in this study are the idealised forest change scenarios and the inactive carbon module in the simulations. Concerning the forest change scenarios, despite efforts to minimise uncertainty by maintaining the current proportions of tree sub-types (i.e., variants of coniferous and broadleaved trees represented in the model), some level of uncertainty remains. It is evident that different tree species have varying suitable growth regions under specific local climate conditions[39,40], and climate change could alter their suitability[41]. Therefore, for more realistic (e.g. species-level) assessments, future studies could incorporate forest change scenarios that consider the suitability of tree species for future local conditions. Regarding the carbon module, the selection of the satellite phenology mode in the model precludes consideration of how forest changes impact the carbon cycle[42]. In reality, changes in forest management can significantly influence the role of forests as carbon sinks[43], a factor that warrants further investigation. Thus, it is crucial to conduct global

simulations that integrate both BGP and BGC impacts, providing more informed guidance for forest management policies.

In addition to the BGC and BGP cooling, other impacts related to forest management should also be considered when designing European forest policies. First, forest ecosystems play a crucial role in maintaining biodiversity, and forestation does not necessarily ensure the restoration of biodiversity. This is related to multiple factors, including negative impacts on plant-pollinator networks[44], competitive disadvantages for local species[45], afforestation-related soil acidification[46], etc. Second, the hydrological consequences of forest management need to be thoroughly understood, as they may exacerbate local and regional water scarcity issues. This can be caused by forestation-induced reduction in runoff[47], increase in evapotranspiration[48,49], or changes in the large-scale water cycle[50–52]. As a result, the resilience of tree species to droughts, whose frequency has increased in recent decades and is projected to increase in the future, even under low or moderate emissions scenarios[53,54], need to be highlighted in European forest policy. Third, the risk of wildfires, particularly in planned afforestation regions, should be thoroughly assessed[55]. In addition to the enhanced probability of wildfires caused by climate change and human activities[56,57], forestation further exacerbates this risk for two main reasons: low resilience[58] and high local fuel load relative to grasslands or croplands[59]. Despite these uncertainties, this study highlights the importance of forest management for local BGP effects over Europe. Science-based decision-making for future forest planning can help mitigate climate change through BGC effects and decrease local impacts due to BGP cooling.

## Methods

### COSMO-CLM$^2$ model

The COSMO-CLM (COnsortium for Small-scale Modelling-Climate Limited-area Modeling Community) regional climate model, coupled with the Community Land Model (COSMO-CLM+CLM: COSMO-CLM$^2$)[60] is used in this study to simulate the climatic feedback to changes in forests. COSMO-CLM is a non-hydrostatic, limited-area atmospheric model, which has been commonly used in regional climate modelling and has shown satisfying performance among regional climate models, especially over Europe[61]. The default land component of COSMO-CLM is the soil module TERRA_ML[62], a simplified land surface scheme that fails to represent the sub-grid cell heterogeneity.

To expand the representation of land surface processes in COSMO-CLM, the Community Land Model (CLM) has been coupled to COSMO-CLM to replace the original TERRA_ML[60]. This coupling was carried out for the first time with version 4 of COSMO and version 3.5 of CLM. Owing to the enhanced implementations of hydrology, biogeophysics, and biogeochemistry of CLM, the coupled model outperforms the uncoupled version in simulating land surface energy fluxes, and then shows a better evaluation result of reproducing temperature and precipitation in Europe[60]. Since the first coupling, both components have been further developed, and in this study, version 6 of COSMO[63] and version 5 of CLM[64] are used.

Community Land Model version 5 (CLM5) is the land component of the Community Earth System Model version 2 (CESM2), which contains a detailed sub-grid cell structure representing different land-use types, including natural vegetation, cropland, urban area, water body, and glacier[64]. Under the land-use tile of natural vegetation, several vegetation types are considered, consisting of three coniferous tree types, five broadleaved tree types, and three grassland types[65]. These different vegetation types vary in parameters regarding their land properties, vegetation growth, photosynthesis, etc., and therefore have various BGP and BGC impacts. After receiving the meteorological forcings from the atmosphere model or external data sets, the land processes are simulated individually over each vegetation type. In this study, the satellite phenology (SP) mode is used, in which the phenology of vegetation is prescribed based on external data sets[66,67],

without the disturbance of other factors. Considering that we focus on the BGP impacts of forest changes, CLM5 with SP mode is an ideal tool for this study.

In CLM5, the 2-meter air temperature ($T_{air}$) is interpolated from surface and atmospheric conditions using Monin-Obukhov similarity theory,

$$T_{air} = T_{sfc} + \left(T_{atm} - T_{sfc}\right) \cdot \frac{\ln\left(\frac{z_{2m}-d}{z_{0h}}\right) - \psi_h\left(\frac{z_{2m}-d}{L}\right)}{\ln\left(\frac{z_{atm}-d}{z_{0h}}\right) - \psi_h\left(\frac{z_{atm}-d}{L}\right)}, \qquad (1)$$

where $T_{sfc}$ is the land surface temperature, $T_{atm}$ is the temperature of the lowest atmosphere level, $z_{0h}$ is the roughness length for sensible heat, $z_{2m}$ equals to 2 meters, d indicates the displacement height (zero for bare soil and nonzero for vegetated canopies), L is the Monin-Obukhov length, $\psi_h$ is the stability correction function for heat, and $z_{atm}$ is the height of atmospheric reference level.

$T_{atm}$ is simulated by the atmosphere model COSMO-CLM, and $T_{sfc}$ is calculated in the land model, CLM, based on the surface energy balance,

$$LW_{up} = SW_{down} - SW_{up} + LW_{down} - LE_{up} - H_{up} - G_{down}, \qquad (2)$$

where $SW_{down}$ and $SW_{up}$ are down- and up-welling shortwave radiation, $LW_{up}$ and $LW_{down}$ are up- and down-welling longwave radiation, $LE_{up}$ indicates latent heat flux from land to the atmosphere, $H_{up}$ indicates sensible heat flux from land to the atmosphere, and $G_{down}$ is the flux from land surface to the ground. $LW_{up}$ is directly determined by $T_{sfc}$,

$$LW_{up} = \varepsilon \sigma T_{sfc}^4 \qquad (3)$$

### Input datasets

The present-day land use and forest types are derived from the Land-Use Harmonisation 2 (LUH2) dataset[28], which provides a reconstruction of historical and future land use based on multiple sources. This dataset has been integrated into the Community Earth System Model version 2 (CESM2), where CLM5 serves as the land component, for simulations in the Coupled Model Intercomparison Project Phase 6 (CMIP6)[68] and related sub-MIPs such as Land-Use MIP[69]. For the purpose of the CMIP6 simulations, the year 2000 is used as a representation of the present day in CESM2, and we adopt this setting in our study.

Given that the SSP5-8.5 scenario is highly unlikely under current climate policies[27], we select SSP3-7.0, a medium-to-high emissions scenario, for this study in order to capture a broad range of realistic warming levels. The simulations from the higher-resolution version of the Max-Planck Institute Earth System Model (MPI-ESM1.2-HR, first realisation)[70,71], part of the CMIP6 initiative, are selected to provide the boundary conditions. This model is chosen because it is considered moderate among both CMIP5 and CMIP6 models, with its equilibrium climate sensitivity and emergent constraints on future warming falling within the range estimated by the IPCC Fifth Assessment Report (AR5)[72].

### Experimental design and outputs analysis

Three sets of forest scenarios are designed in this study, representing forestation/deforestation, forest species conversion, and the combination of forestation and forest species conversion. Together with the control simulation (Ctl) based on the present-day natural vegetation distribution, we conduct seven simulations in total (Table 1). These experiments only vary in the natural vegetation distribution (coniferous forests, broadleaf forests, and grassland; other land-use types, such as cropland, remain identical in all experiments). To assess the sensitivity of BGP climate impacts to canopy height and leaf area index

(LAI), we also design six extra scenarios changing monthly LAI (LAI⁺: LAI multiplied by 1.5; LAI⁻: LAI divided by 1.5) or canopy height (HGT⁺: canopy heights multiplied by 1.5; HGT⁻: canopy heights divided by 1.5) of natural vegetation or both (L⁺H⁺: LAI and canopy heights both multiplied by 1.5; L⁻H⁻: LAI and canopy heights both divided by 1.5). The analysis of these additional experiments is presented in the Supplementary Note 2.

Simulations of forest coverage and forest composition start from the year 2015 and end in the year 2059, with the first 10 years as a spin-up period, so the 35-year period 2025-2059 is used for analysis. Simulations of forest health end in the year 2034, and the period 2020-2034 is used for analysis. Simulation resolution is 0.44° × 0.44° for the atmosphere and 0.5° × 0.5° for the land model. Outputs include monthly mean, daily maximum, and daily minimum temperature, and monthly mean energy fluxes (analysed in Supplementary Note 2).

## Data availability

The data generated for this study have been deposited in the figshare database with the license CC BY 4.0: https://figshare.com/articles/dataset/Yao_et_al_2025_Conversion_from_coniferous_to_broadleaved_trees_can_make_European_forests_more_climate-effective/29995021?file=57457759[73]. The raw outputs of simulations can be obtained by inquiring the corresponding author.

## Code availability

COSMO − CLM² is not publicly accessible. All scripts developed for this study are available at: https://github.com/YiYao1995/Yao-et-al-2025_Conversion_from_coniferous_to_broadleaved.git[74].

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

## Acknowledgements
Y.Y., F.J. and S.I.S. acknowledge funding by the EU Horizon Europe project ForestNavigator—Navigating European forests and forest bioeconomy sustainably to EU climate neutrality (GA No. 101056875). S.J.D.H. acknowledges funding by the Belgian Federal Science Policy Office BELSPO (B2/223/P1/DAMOCO and SR/00/410/AFROCARDS).

## Author contributions
Y.Y., P.S., J.S., M.R., J.P., and S.I.S. designed the study. Y.Y. wrote the manuscript with support from P.S., M.H., J.S., F.J., F.B., M.R., J.P., M.W., A.L.D.A., S.J.D.H., V.G., M.G.W., J.G., A.C., F.D.F., P.H., and S.I.S., and performed all analyses under the supervision of P.S., M.W., J.G., and S.I.S. Y.Y., P.S., J.S., A.L.D.A., A.C., F.D.F., P.H., and S.I.S. participated in the protocol design of COSMO-CLM$^2$ simulations. F.J., F.B., and S.J.D.H. helped with the preparation of input datasets for COSMO-CLM$^2$ simulations. Y.Y., P.S., and M.H. performed COSMO-CLM$^2$ simulations. M.W. provided guidance on the analysis of forest-induced impacts on energy fluxes. V.G. provided guidance on the analysis of changes in European forests. M.G.W. provided valuable insights regarding other parts of forestation-induced impacts. A.C. provided valuable insights regarding the additional considerations of forest policy.

## Funding

## Competing interests
The authors declare no competing interests.
