## [Transparent Peer Review file · Nature Communications]

Conversion from coniferous to broadleaved trees can make European forests more climate-effective

Corresponding Author: Dr Yi Yao

Version 0:

Reviewer comments:

Reviewer #1

(Remarks to the Author)

Review of the manuscript

“Conversion from coniferous to broadleaved trees can make European forests more climate-effective”

by

Yao et al.

for

General remarks:

This study examines the potential of broadleaved trees to mitigate summer hot extremes in Europe. For this purpose, regional climate simulations with COSMO-CLM² are performed for the period 2025-2059 for different forestation scenarios. The results show that a change in the current composition of European forests from coniferous to broadleaved trees would reduce the monthly mean daily maximum temperatures in summer by 0.6 K. Moreover, a tailored afforestation with broadleaved trees would reduce summer hot extremes by 0.7 K, while afforestation with the current forest composition would result in a warming of 0.3 K.

The paper is clearly structured, comprehensibly written and fits well with the scope of Nature Communications. However, there are some issues which need to be addressed by the authors before the manuscript is ready for publication.

Major comments:

1. It is a bit difficult for me to interpret the results regarding the effect of Brd versus Def. For example: T_n in summer is higher in Def than in Brd, which is the atmospheric response to afforestation that one would expect. T_x in Def is also slightly higher than in Brd, but the differences are smaller. On the Iberian Peninsula, Brd even has a warming effect, and in north-eastern Europe at least not a cooling effect. The daily mean values, on the other hand, show a dipole with warmer temperatures in northern Europe and colder temperatures in the south in Brd. This doesn't fit together for me and I think it has to do with the use of the 2m temperature. Afforestation can have opposite effects at the surface and in the atmosphere. Using the 2m as a weighted mean of T_s and T_a can therefore be problematic. In this example, for instance, it looks to me as if Brd only has a slight cooling effect in the atmosphere during the day (in northern Europe even a warming effect), but this is not visible in T_{2m} by averaging with T_s , as a cooling occurs at T_s . However, since the daily mean temperature also considers the transitions between the potential warming effect in T_a during the day and the cooling effect at night, different patterns can be seen for T_{2m} than for T_x and T_n . It would help if this could be checked and, if necessary, taken into account in the evaluation of the effects of Brd versus Def.

2. The paper shows regions in which afforestation with broadleaved trees has a cooling effect and in which it does not. But why this is the case in a certain region is not really explained. I am aware that an in-depth analysis of the results is not provided for in this journal, and more is packed into the appendix. However, as this study are also intended to serve as a basis for decision-makers, this information would be very important (one or two sentences in the paper would be sufficient). Because I think the regions in which afforestation has a cooling effect are model-dependent, as for example shown in Davin et al., (2020). However, if the processes that lead to a warming or a cooling effect are briefly described, indicators for these processes can be searched in nature and decisions can thus be adapted.

Minor comments:

Line 30: "that the Northern and Central regions should be prioritised for forestation"

- Do you mean in general or for a change to broadleaved trees? As far as I can see, this is a result that has not been seen in previous studies (e.g. Schwaab et al., 2020, Breil et al., 2023). Should therefore be discussed.

Line 38: "the BGP impacts of land cover changes are relatively small compared to the BGC impacts on the global scale"

- Does this refer to historical land use changes or to the BGP potential in general? I doubt that changes in the albedo for instance do not have a strong effect on the energy balance, especially in the high latitudes in connection with the snow masking effect. Sure, some of the additional absorbed radiation can be re-emitted through increased longwave radiation, but there will still be more energy in the system that will affect the global climate through remote effects.

Line 51: "converting a large amount of energy into latent heat"

- Not only that. In the lower latitudes, cloud feedback is also particularly important, which counteracts the lower albedo. Or do you mean here the lower latitudes in Europe?

Line 114: "Regarding daily minimum temperatures (TN), the cooling impacts are less pronounced, and Def contributes to the warming effects in the central and southern parts of Europe, particularly on TN,JJA (Figure A3)"

- See major comment 1

Line 123: "In the Northern region (Figure 3c, A4c, and A5c), Def results in reductions of TmonX, TmonM, and TmonN by approximately 2.8°C, 1.9 °C, and 1.3 °C, respectively, in April and May"

- It would be good if you briefly explain that this is caused by the snow masking effect as described in the supplement (see major comment 2). Because this could mean that this cooling effect will decrease in more far future in the context of climate change and reduced snow covers.

Line 157: "An exception is observed in the Southern region for TmonX, where AfB induces a warming effect in most months, and its cooling potential in summer is lower than both Def and Brd"

- Why is this the case (see major comment 2)? If several regions are not suitable for forest management it would be good to know why as a basis for decision-making. Is it caused by soil moisture limitation?

Line 205: "so policies may be needed to incentivise foresters to shift from commercially valuable species to climate-friendly alternatives."

- the question is whether conifers will still be as economically attractive in the future as they are today. Conifers are generally shallow-rooted, which is likely to lead to drought stress and lower yields as a result of climate change and the associated increased drought frequency.

Supplementary Note 2, page 11: "However, in summer and on yearly averages (Figure S7b,f), these effects are reverse to cooling impacts due to increased LHF (Figure S9f) from higher LAI (Figure S3b). Although SHF is increased (Figure S10f), the decreased LWup still causes a lower TsumM."

- I'm not sure if this is the case here. It is possible that the reduced long-wave radiation has a stronger effect on the near-surface temperatures than an increased sensible heat flux. But it could also be an effect of the analyzed 2m temperature. As mentioned in major comment 1, afforestation can cause opposite temperature responses at the surface and in the atmosphere. And since the 2m is a weighted mean between T_s and T_a , temperature changes are more difficult to interpret (e.g. Winckler et al., 2019, Breil et al., 2020). Here, however, the differences between surface and atmosphere do not seem to be so contradictory. In both T_{2m} and LW_{up} , which can be interpreted as a pattern of T_s according to Stefan-Boltzmann, there is a dipole, with warming in the north and cooling in the south (but much more pronounced in T_s) in Aff. It is intuitive that the increased SW_{net} in Aff leads to warming in the north, but the question remains why there is cooling in the south? This can happen if, due to the properties of trees (increased LAI, increased roughness...), more energy is released into the atmosphere (sum of the turbulent fluxes) than is additionally absorbed despite increased SW_{net} . A generally increased sensible heat flux (SH) warms the atmosphere despite lower T_s , which is the reason why, for example, the T_{2m} in Great Britain is not reduced, although it is T_s .

The regions in which T_{2m} is most strongly reduced are therefore the regions in which SH is reduced or not increased and thus the atmosphere is not additionally warmed, as in the Balkans or northern France. The reduced SH there seems to be due to a non-increased SW_{net} (perhaps due to increased cloud cover?). So the energy amount at the surface is not increased. However, since the properties of the trees allow a greater proportion of the available energy to be converted into LH, less remains for SH.

Supplementary Note 2, page 11: "This indicates that in spring, the energy influxes exceed the outfluxes, and vice versa in summer, which could be related to the growth of vegetation (Figure S2)."

- That's interesting, but I don't quite understand it. How does this relate to vegetation growth? A reduced residual of the surface energy balance with afforestation in summer in northern Europe should actually mean that more energy is released than is additionally absorbed, which in turn should lead to cooling. However, it is getting warmer in the region. Does the model simulate heat storage in the vegetation?

Supplementary Figure S17:

- The labels in the figure are incorrect.

References:

- Breil, M., and Coauthors, (2020). The opposing effects of reforestation and afforestation on the diurnal temperature cycle at the surface and in the lowest atmospheric model level in the European summer. *Journal of Climate*, 33(21), 9159-9179.
- Breil, M., Weber, A., & Pinto, J. G. (2023). The potential of an increased deciduous forest fraction to mitigate the effects of heat extremes in Europe. *Biogeosciences*, 20(12), 2237-2250.
- Schwaab, J., Davin, E. L., Bebi, P., Duguay-Tetzlaff, A., Waser, L. T., Haeni, M., & Meier, R. (2020). Increasing the broad-leaved tree fraction in European forests mitigates hot temperature extremes. *Scientific reports*, 10(1), 14153.
- Winckler, J., and Coauthors, (2019): Different response of surface temperature and air temperature to deforestation in climate models. *Earth Syst. Dyn.*, 10, 473–484.

Reviewer #2

(Remarks to the Author)

Dear authors and editors,

European forests play a significant role in climate policy frameworks. However, afforestation may result in a warming effect due to increased absorption of solar radiation, potentially outweighing the cooling influence of evapotranspiration. This study employs regional climate models to simulate various forest change scenarios, with the objective of investigating strategies to maximize the climate mitigation potential of forests in the context of global warming. The topic is highly relevant and holds substantial practical significance. However, the following issues remain to be further clarified and refined.

1. The authors illustrate the spatial variations of several key variables, including leaf area index (LAI), sensible heat, latent heat, albedo, and surface roughness, across different experimental scenarios. However, there is a lack of clear quantification and detailed discussion regarding the contributions of these variable variations in different regions to changes in extreme temperatures. It is suggested that a schematic diagram be included to help readers better understand the underlying mechanisms through a more intuitive and comprehensive visual representation. It is likely that the dominant factors differ across regions.

2. What was the rationale for selecting the SSP3-7.0 scenario? Would the conclusions differ under alternative scenarios (e.g., SSP5-8.5)?

3. How can cropland be treated? Does it be treated as grassland in various experiments? Or do the analysis focus exclusively on areas covered by natural vegetation?

4. L115-118: Why do the Brd and Def simulations exhibit a significant cooling effect on T_M (Figure A1) and T_X (Figure A2), yet demonstrate only a limited impact on T_N (Figure A3)—particularly during summer—and in some cases, why does the Def test even lead to widespread warming (Figure A3g)? Please provide a detailed explanation of the underlying factors contributing to these differences.

5. Each species occupies a specific ecological niche. If climatic and environmental conditions fail to meet the species' requirements, its survival, reproduction, and expansion may be significantly constrained. As illustrated in Figure 1, the dominant vegetation type in the study area is coniferous forest, with relatively low coverage of broad-leaved forest. Therefore, do the Brd and AfB experiments possess practical guiding significance?

Minor comments:

- L24: the sentence “e.g.,, the monthly mean daily maximum temperature.....” has an extra comma.
- L148: “This warming effect does not exit on T_M , JJA and T_N , JJA (Figure A7 and Figure A8)” should be modified as “..... (Figure A6 and Figure A8)”?
- Adjust the vertical axis range of certain graphs (e.g., Figure S23) to facilitate a clearer visualization of variable changes for readers.
- Some variable indices in the figures contain inaccuracies, such as in Figure S17 d-i. Other diagrams also exhibit similar issues. A thorough review is recommended.

Version 1:

Reviewer comments:

Reviewer #1

(Remarks to the Author)

The authors have addressed all my concerns, therefore, I recommend to accept the manuscript.

(Remarks on code availability)

Reviewer #2

(Remarks to the Author)

The authors have provided detailed explanations and elaborations on my scientific comments, and have further improved the presentation of the article. This is a meaningful study and is recommended for publication.

(Remarks on code availability)

All the numerical simulation data in the article are involved, and is accompanied by a README file for instructions.

1 Reviewer 1

1.1 Major comments

Reviewer 1 Comment 1

This study examines the potential of broadleaved trees to mitigate summer hot extremes in Europe. For this purpose, regional climate simulations with COSMO-CLM² are performed for the period 2025-2059 for different forestation scenarios. The results show that a change in the current composition of European forests from coniferous to broadleaved trees would reduce the monthly mean daily maximum temperatures in summer by 0.6 K. Moreover, a tailored afforestation with broadleaved trees would reduce summer hot extremes by 0.7 K, while afforestation with the current forest composition would result in a warming of 0.3 K. The paper is clearly structured, comprehensibly written and fits well with the scope of Nature Communications. However, there are some issues which need to be addressed by the authors before the manuscript is ready for publication.

Response We thank Reviewer 1 for those supportive words and constructive suggestions, which helped us to improve the manuscript. Below, we address each comment carefully and explain the corresponding changes in the revised manuscript accordingly.

Reviewer 1 Comment 2

It is a bit difficult for me to interpret the results regarding the effect of Brd versus Def. For example: T_n in summer is higher in Def than in Brd, which is the atmospheric response to afforestation that one would expect. T_x in Def is also slightly higher than in Brd, but the differences are smaller. On the Iberian Peninsula, Brd even has a warming effect, and in north-eastern Europe at least not a cooling effect. The daily mean values, on the other hand, show a dipole with warmer temperatures in northern Europe and colder temperatures in the south in Brd. This doesn't fit together for me and I think it has to do with the use of the 2m temperature. Afforestation can have opposite effects at the surface and in the atmosphere. Using the 2m as a weighted mean of T_s and T_a can therefore be problematic. In this example, for instance, it looks to me as if Brd only has a slight cooling effect in the atmosphere during the day (in northern Europe even a warming effect), but this is not visible in T_{2m} by averaging with T_s , as a cooling occurs at T_s . However, since the daily mean temperature also considers the transitions between the potential warming effect in T_a during the day and the cooling effect at night, different patterns can be seen for T_{2m} than for T_x and T_n . It would help if this could be checked and, if necessary, taken into account in the evaluation of the effects of Brd versus Def.

Response We thank Reviewer 1 for this suggestion. Indeed, in the model we used, the 2-meter air temperature is not a prognostic variable but rather a diagnostic one, derived through interpolation from the surface skin temperature (T_{sfc}) and the lowest atmosphere level temperature (T_{atm}). Thus, to address the concerns raised here by the reviewer, we also analysed T_{sfc} and T_{atm} and added the results to the main manuscript. In addition, we added some text to the Introduction section with citations to previous studies, which emphasises

this issue.

Introduction, Lines 87-90

In addition, near-surface air temperature was commonly used in previous research, but some studies revealed that forest changes-induced impacts on different temperature variables (like surface skin temperature and 2-meter air temperature) may vary, emphasising the need to consider the entire near-surface temperature profile (1; 2).

Introduction, Lines 99-101

We also examine the impacts of forest changes on the temperatures of the land surface and the lowest atmosphere model level, and analyse surface energy fluxes to identify the drivers of the changes in temperatures.

Results, Lines 126-142

Although both Brd and Def reduce the summer mean SW_{net} , changes in summer mean daily maximum surface skin temperature (T_{sfc} , directly determining upwelling long-wave radiation, LW_{up}) under the two scenarios exhibit opposite patterns, with Brd leading to cooling and Def resulting in warming across most of the study area (FIG 1b,c), possibly due to differences in the partitioning of land surface energy fluxes. Both Brd and Def reduce the summer mean sensible heat flux from the land to the atmosphere (H_{up} ; Figure S2j,k) as a result of decreased summer mean SW_{net} . However, Brd enhances the summer mean latent heat flux (LE_{up}) due to the higher evapotranspiration rate of broadleaf trees (Figure S2b), whereas Def leads to a reduction in summer mean LE_{up} across most of Europe (Figure S2c). As a result, more energy at the land surface is released as summer mean LW_{up} under Def (Figure S2s), and at the same time, a decrease in summer mean H_{up} (Figure S2k) indicates a lower ability of the land surface to heat up the air, which leads to a decrease in summer mean daily maximum temperature of the lowest atmosphere level (T_{atm}). Different patterns emerge when considering daily mean and daily minimum temperatures (Figure S3,S4), such as in Southern Europe, both summer mean daily minimum T_{air} and T_{atm} increase under Def (Figure S4g,k). This may be explained by an increase in nighttime H_{up} resulting from reduced LE_{up} under Def, which becomes the dominant factor in the absence of shortwave radiation.

Results, Lines 177-189

When examining changes in the monthly mean daily maximum, mean, and minimum T_{sfc} , T_{air} , and T_{atm} , distinct seasonal patterns emerge under Def (FIG 2, S5, and S6). In general, changes in monthly mean daily maximum T_{air} and T_{atm} are closely aligned, whereas T_{sfc} exhibits markedly different behaviour. For instance, over the Atlantic region, monthly mean daily maximum T_{sfc} increases by approximately 0.75 C in August, while both monthly mean daily maximum T_{air} and T_{atm} show slight decreases (FIG 2f). In contrast, under Brd, the seasonal patterns of monthly mean daily maximum T_{sfc} remain broadly consistent with those of monthly mean daily maximum T_{air} and T_{atm} , despite some differences in magnitude. Similar patterns are also observed for the mean and min-

imum temperatures (Figures S5 and S6). As discussed earlier, these differences are closely linked to the partitioning of surface energy fluxes, with the majority of the energy out-flux being released as upwelling longwave radiation (LW_{up} ; Figure S7). More specifically, between April and October, Brd facilitates a greater magnitude of the latent heat flux (LE_{up}), whereas this is reduced in Def, inducing higher LW_{up} .

Results, Lines 201-206

Similar to Def, Aff has opposite impacts on summer mean daily maximum T_{air} and T_{sfc} , with the cooling impact on T_{sfc} exceeding 1.0 or even 2.0 C in some grid cells (FIG 3a,d). More specifically, compared to grassland, forest can absorb more shortwave radiation, but can also increase both turbulent fluxes (summer mean LE_{up} and H_{up}), leading to a decrease in LW_{up} (Figure S9a,i). However, increased H_{up} substantially heats up the air

Results, Lines 242-245

FIG 1: Conversion from conifer to broadleaf trees in existing forests can mitigate summer hot extremes more than deforestation. Multi-year (2025-2059, in the experiment Ctl) summer (June, July, and August) mean daily maximum land surface temperature ($T_{sfc,jjaX}$: a), 2-meter air temperature ($T_{air,jjaX}$: e), and the temperature at the lowest atmospheric level ($T_{atm,jjaX}$: i). Changes in these temperatures, respectively, under the conversion from conifer to broadleaf forests scenario (Brd-Ctl: b,f,j), by the deforestation scenario (Def-Ctl: c,g,k), and the difference between the two scenarios (Brd-Def: d,h,l).

Moreover, A_{ff} leads to substantial spring warming of monthly mean daily maximum T_{sfc} in Alpine and Northern regions, which can be partially offset by A_{fB} due to a decrease in monthly mean SW_{net} (Figure S14a–d).

FIG 2: Different seasonal patterns of biogeophysical impacts of forest composition change and deforestation across five regions. Changes in multi-year mean (2025-2059) monthly mean daily maximum temperature (T_{monX}) induced by the conversion from conifer to broadleaf forests scenario (Brd) (left column) and the deforestation scenario (Def) (right column). The values shown are the regionally averaged temperature change (minus the control simulation) over five regions: Alpine (a-b), Northern (c-d), Atlantic (e-f), Continental (g-h), and Southern (i-j).

FIG 3: Forest composition conversion can mitigate forestation-induced warming. Changes in multi-year (2025-2059, compared to the experiment Ctl) summer (June, July, and August) mean daily maximum land surface temperature ($T_{sfc,jjaX}$: **a-d**), 2-meter air temperature ($T_{air,jjaX}$: **e-h**), and the temperature at the lowest atmospheric level ($T_{atm,jjaX}$: **i-l**) under the forestation scenario (Aff-Ctl: **a,e,i**), by the combining forestation and conversion from coniferous to broadleaf forests scenario (AfB-Ctl: **b,f,j**), the difference between the two scenarios (AfB-Aff: **c,g,k**), and the difference between AfB and the deforestation scenario (AfB-Def: **d,h,l**).

FIG 4: Spatially variant seasonal pattern of forestation-induced biogeophysical impacts across five regions. Changes in multi-year mean (2025-2059) monthly mean daily maximum temperature (T_{monX}) induced by the forestation scenario (Aff-Ctl) (**left column**) and the combining scenario of forestation and the conversion from coniferous to broadleaf forests (Afb-Ctl) (**right column**). The values shown are the regionally averaged temperature change (minus the outputs from the Ctl simulation) over five regions: Alpine (**a-b**), Northern (**c-d**), Atlantic (**e-f**), Continental (**g-h**), and Southern (**i-j**) (see Figure 1e).

We also added and described in the Method section how the 2-meter air temperature is diagnosed in the CLM5.

Methods, Lines 392-399

In CLM5, the 2-meter air temperature (T_{air}) is interpolated from surface and atmospheric conditions using Monin–Obukhov similarity theory,

$$T_{\text{air}} = T_{\text{sfc}} + (T_{\text{atm}} - T_{\text{sfc}}) \cdot \frac{\ln\left(\frac{z_{2\text{m}}-d}{z_{0\text{h}}}\right) - \psi_h\left(\frac{z_{2\text{m}}-d}{L}\right)}{\ln\left(\frac{z_{\text{atm}}-d}{z_{0\text{h}}}\right) - \psi_h\left(\frac{z_{\text{atm}}-d}{L}\right)} \quad (1)$$

, where T_{sfc} is the land surface temperature, T_{atm} is the temperature of the lowest atmosphere level, $z_{0\text{h}}$ is the roughness length for sensible heat, $z_{2\text{m}}$ equals to 2 meters, d indicates the displacement height (zero for bare soil and nonzero for vegetated canopies), L is the Monin–Obukhov length, ψ_h is the stability correction function for heat, and z_{atm} is the height of atmospheric reference level.

Reviewer 1 Comment 3

The paper shows regions in which afforestation with broadleaved trees has a cooling effect and in which it does not. But why this is the case in a certain region is not really explained. I am aware that an in-depth analysis of the results is not provided for in this journal, and more is packed into the appendix. However, as this study are also intended to serve as a basis for decision-makers, this information would be very important (one or two sentences in the paper would be sufficient). Because I think the regions in which afforestation has a cooling effect are model-dependent, as for example shown in Davin et al., (2020). However, if the processes that lead to a warming or a cooling effect are briefly described, indicators for these processes can be searched in nature and decisions can thus be adapted.

Response We thank Reviewer 1 for this suggestion. We fully agree that a more comprehensive analysis is necessary. We therefore added some analysis to the revised main manuscript version. Unfortunately, the model did not output sub-daily energy fluxes, so it is difficult to attribute changes in daily maximum and minimum temperatures. In addition, we added one panel in the figures to compare the results between AfB and Def to directly investigate the impacts of the conversion from grassland to broadleaf forests.

Results, Lines 118-124

However, in the Mediterranean region, compared to Def, Brd has less cooling effect, whose effect can exceed 0.5 C in some grid cells (FIG 1e). The cooling effects observed in both scenarios are strongly associated with reductions in summer mean net shortwave radiation (SW_{net} ; Figure S1j,k), primarily driven by increases in summer mean upwelling shortwave radiation (SW_{up} ; Figure S1b,c). Additionally, Brd causes a slight decrease in summer mean downwelling shortwave radiation (SW_{down} ; Figure S1f), whereas Def leads to an increase in many regions, potentially due to changes in cloud cover (Figure S1g).

1.2 Minor comments

Reviewer 1 Comment 4

Line 30: “that the Northern and Central regions should be prioritised for forestation” - Do you mean in general or for a change to broadleaved trees? As far as I can see, this is a result that has not been seen in previous studies (e.g. Schwaab et al., 2020, Breil et al., 2023). Should therefore be discussed.

Response We are sorry for the confusion, and we perfectly agreed with the reviewer. The key point we wish to emphasise is that forestation, along with the conversion of coniferous forests to broadleaf forests, should be recognized and evaluated as a potential strategy for mitigating heat extremes. Considering that in the Southern regions there are warming effects, and the effects in Alpine regions are small, we concluded that Northern and Central regions should be prioritised. We added some text to the Discussion section to clarify and emphasise this point.

Discussion, Lines 262-267

Another modelling-based study (3) also confirmed that converting coniferous to deciduous forests can effectively reduce the intensity of heat extremes over several grid cells in Europe, but this effect is simulated to be minor in Scandinavian regions. This spatial variance aligns with an observationally-based study (4) but differs from the results of the present work. The discrepancy may stem from differences in study periods, as the background climate in this study is warmer, potentially amplifying the evaporative cooling associated with forest-type changes.

Reviewer 1 Comment 5

Line 38: “the BGP impacts of land cover changes are relatively small compared to the BGC impacts on the global scale” - Does this refer to historical land use changes or to the BGP potential in general? I doubt that changes in the albedo for instance do not have a strong effect on the energy balance, especially in the high latitudes in connection with the snow masking effect. Sure, some of the additional absorbed radiation can be re-emitted through increased longwave radiation, but there will still be more energy in the system that will affect the global climate through remote effects.

Response We thank Reviewer 1 for pointing this out. Yes, this statement was made based on studies investigating BGP and BGC impacts of historical anthropogenic land cover change. We have therefore adjusted this sentence.

Introduction, Lines 40-43

Although the BGP impacts of land cover changes of historical anthropogenic land cover change are relatively small compared to the BGC impacts on the global scale, they can substantially affect and even dominate the local climate pattern in some regions (5; 6; 7).

Reviewer 1 Comment 6

Line 51: “converting a large amount of energy into latent heat” - Not only that. In the lower latitudes, cloud feedback is also particularly important, which counteracts the lower albedo. Or do you mean here the lower latitudes in Europe?

Response We thank Reviewer 1 for pointing this out. We adjusted this sentence (see below).

Introduction, Lines 52-56

More specifically, in low-latitude regions with ample water availability, forestation can substantially increase local evapotranspiration, converting a large amount of energy into latent heat. Conversely, in high-latitude regions, most of the additionally absorbed solar radiation becomes sensible heat and upwelling longwave radiation.

Reviewer 1 Comment 7

Line 114: “Regarding daily minimum temperatures (TN), the cooling impacts are less pronounced, and Def contributes to the warming effects in the central and southern parts of Europe, particularly on TN,JJA (Figure A3)” - See major comment 1

Response Please check Reviewer 1 Comment 2. Specifically, regarding the daily minimum temperature change induced by Def, we added some text.

Results, Lines 138-142

Different patterns emerge when considering daily mean and daily minimum temperatures (Figure S3,S4), such as in Southern Europe, both summer mean daily minimum \$T_{\text{air}}\$ and \$T_{\text{atm}}\$ increase under Def (Figure S4g,k). This may be explained by an increase in nighttime \$H_{\text{up}}\$ resulting from reduced \$LE_{\text{up}}\$ under Def, which becomes the dominant factor in the absence of shortwave radiation.

Reviewer 1 Comment 8

Line 123: “In the Northern region (Figure 3c, A4c, and A5c), Def results in reductions of T_{monX} , T_{monM} , and T_{monN} by approximately 2.8°C, 1.9 °C, and 1.3 °C, respectively, in April and May” - It would be good if you briefly explain that this is caused by the snow masking effect as described in the supplement (see major comment 2). Because this could mean that this cooling effect will decrease in more far future in the context of climate change and reduced snow covers.

Response We thank Reviewer 1 for this suggestion and have added some text to this sentence.

Results, Lines 164-167

Def results in reductions of monthly mean daily maximum, mean, and minimum T_{air} by approximately 2.8C, 1.9 C, and 1.3 C, respectively, in April and

May. This substantial cooling may be related to the snow-radiation feedback, which may decrease in a warming world.

Reviewer 1 Comment 9

Line 157: “An exception is observed in the Southern region for TmonX, where AfB induces a warming effect in most months, and its cooling potential in summer is lower than both Def and Brd” - Why is this the case (see major comment 2)? If several regions are not suitable for forest management it would be good to know why as a basis for decision-making. Is it caused by soil moisture limitation?

Response We thank Reviewer 1 for this suggestion. We kindly ask the reviewer to check Reviewer 1 Comment 3.

Reviewer 1 Comment 10

Line 205: “so policies may be needed to incentivise foresters to shift from commercially valuable species to climate-friendly alternatives.” - the question is whether conifers will still be as economically attractive in the future as they are today. Conifers are generally shallow-rooted, which is likely to lead to drought stress and lower yields as a result of climate change and the associated increased drought frequency.

Response We thank Reviewer 1 for pointing this out and therefore have included an additional sentence here reflecting this concern.

Discussion, Lines 333-335

As a result, the resilience of tree species to droughts, whose frequency has increased in recent decades and is projected to increase in the future, even under low or moderate emissions scenarios (8; 9).

Reviewer 1 Comment 11

Supplementary Note 2, page 11: “However, in summer and on yearly averages (Figure S7b,f), these effects are reverse to cooling impacts due to increased LHF (Figure S9f) from higher LAI (Figure S3b). Although SHF is increased (Figure S10f), the decreased LWup still causes a lower TsumM.”

- I’m not sure if this is the case here. It is possible that the reduced long-wave radiation has a stronger effect on the near-surface temperatures than an increased sensible heat flux. But it could also be an effect of the analyzed 2m temperature. As mentioned in major comment 1, afforestation can cause opposite temperature responses at the surface and in the atmosphere. And since the 2m is a weighted mean between Ts and Ta, temperature changes are more difficult to interpret (e.g. Winckler et al., 2019, Breil et al., 2020). Here, however, the differences between surface and atmosphere do not seem to be so contradictory. In both T_{2m} and LWup, which can be interpreted as a pattern of Ts according to Stefan-Boltzmann, there is a dipole, with warming in the north and cooling in the south (but much more pronounced in Ts) in Aff. It is intuitive that the increased SWnet in Aff leads to warming in the north, but the question remains why there is cooling in the south? This can happen if, due to the properties of trees (increased LAI, increased roughness...), more energy is released into the atmosphere (sum of the turbulent fluxes) than is additionally absorbed despite increased SWnet. A generally increased sensible heat flux (SH) warms the atmosphere despite lower Ts, which is the reason why, for example, the T_{2m} in Great Britain is not reduced, although it is Ts. The regions in which T_{2m} is most strongly reduced are therefore the regions in which SH is reduced or not increased and thus the atmosphere is not additionally warmed, as in the Balkans or northern France. The reduced SH there seems to be due to a non-increased SWnet (perhaps due to increased cloud cover?). So the energy amount at the surface is not increased. However, since the properties of the trees allow a greater proportion of the available energy to be converted into LH, less remains for SH.

Response We thank Reviewer 1 for raising this concern. Regarding this point, we kindly ask the reviewer to check Reviewer 1 Comment 2.

Reviewer 1 Comment 12

Supplementary Note 2, page 11: “This indicates that in spring, the energy influxes exceed the outfluxes, and vice versa in summer, which could be related to the growth of vegetation (Figure S2).” - That’s interesting, but I don’t quite understand it. How does this relate to vegetation growth? A reduced residual of the surface energy balance with afforestation in summer in northern Europe should actually mean that more energy is released than is additionally absorbed, which in turn should lead to cooling. However, it is getting warmer in the region. Does the model simulate heat storage in the vegetation?

Response We agree with the reviewer. Indeed, there may be something which is incorrect. Considering the surface energy balance:

$$LW_{up} = SW_{down} - SW_{up} + LW_{down} - LE_{up} - H_{up} - G_{down} \quad (2)$$

When G_{down} increases, it indicates that $SW_{\text{down}} + LW_{\text{down}} - SW_{\text{up}} - LE_{\text{up}} - H_{\text{up}} - LW_{\text{up}}$ increases, so more correctly, the energy influx from the atmosphere to the land exceed the energy outflux from the land to the atmosphere. Therefore, we re-checked the technical note of the model and found that the increased G_{down} indicates more energy transfer from the land surface to soil layers, so the statement regarding vegetation growth is actually wrong. We apologise for this mistake and have removed this part from the Supplementary Information.

Reviewer 1 Comment 13

Supplementary Figure S17: - The labels in the figure are incorrect.

Response We thank Reviewer 1 for finding this error. We have re-structured the Supplementary Information and we checked through the labels in the figures.

2 Reviewer 2

2.1 Major comments

Reviewer 2 Comment 1

European forests play a significant role in climate policy frameworks. However, afforestation may result in a warming effect due to increased absorption of solar radiation, potentially outweighing the cooling influence of evapotranspiration. This study employs regional climate models to simulate various forest change scenarios, with the objective of investigating strategies to maximize the climate mitigation potential of forests in the context of global warming. The topic is highly relevant and holds substantial practical significance. However, the following issues remain to be further clarified and refined.

Response We sincerely thank Reviewer 2 for the encouraging feedback and valuable suggestions, which have contributed to improving our manuscript. Below, we provide detailed responses to each comment.

Reviewer 2 Comment 2

The authors illustrate the spatial variations of several key variables, including leaf area index (LAI), sensible heat, latent heat, albedo, and surface roughness, across different experimental scenarios. However, there is a lack of clear quantification and detailed discussion regarding the contributions of these variable variations in different regions to changes in extreme temperatures. It is suggested that a schematic diagram be included to help readers better understand the underlying mechanisms through a more intuitive and comprehensive visual representation. It is likely that the dominant factors differ across regions.

Response We thank Reviewer 2 for this valuable suggestion. In response, we have added a schematic diagram (FIG 5) and tables (TAB 1–5) in the revised manuscript to illustrate the changes in energy fluxes and temperatures. We hope that these additions will help readers

better understand the mechanisms underlying the effects of changes in forest composition, as well as their magnitude.

Results, Lines 144-159

To better quantify and understand the mechanisms underlying the impacts of forest changes on temperature, we calculate the summer mean energy fluxes and temperatures over five sub-regions (FIG 5a-c). For example, in the Atlantic region, both Brd and Def increase summer mean SW_{up} from 25.82 to 28.40 and 29.99 $W m^{-2}$, respectively, due to the albedo increase associated with the conversion of coniferous forests to broadleaf forests or grassland. In contrast, Brd slightly decreases summer mean SW_{down} by 185.58 to 184.71 $W m^{-2}$, whereas Def increases it from 185.58 to 187.32 $W m^{-2}$, likely owing to changes in cloud cover. Consequently, Brd results in a slightly lower summer mean SW_{net} than Def in summer. Both scenarios have very small effects on summer mean LW_{up} and ground flux (G_{down}), so the increase in summer mean SW_{up} is redistributed among LE_{up} , H_{up} , and LW_{up} . Under Brd, summer mean LE_{up} increases substantially as broadleaf trees generally consume more water, leading to decreases in both H_{up} and LW_{up} , thereby producing cooling effects on all three temperatures. In contrast, under Def, both summer mean LE_{up} and H_{up} decrease, and LW_{up} is therefore enhanced. As a result, summer mean daily maximum T_{sfc} increases, whereas the other two temperatures decrease. Slight differences in these temperature responses may occur when examining other temperature metrics, which can be attributed to the diurnal variability in energy fluxes.

Results, Lines 226-237

Across the Atlantic region, Aff reduces both summer mean SW_{down} (185.58 to 183.30 $W m^{-2}$, possibly due to increased cloud cover) and summer mean SW_{up} (25.82 to 21.25 $W m^{-2}$, likely resulting from an increase in albedo), thereby decreasing SW_{net} (FIG 5a,d). Concurrently, summer mean LE_{up} and H_{up} both increase, contributing to a reduction in LW_{up} . This combination lowers summer mean daily maximum T_{sfc} while increasing T_{atm} and T_{air} . In contrast, AfB markedly decreases summer mean SW_{down} (185.58 to 181.47 $W m^{-2}$, likely due to a substantial increase in cloud cover associated with enhanced evapotranspiration) and slightly increases SW_{up} (25.82 to 26.34 $W m^{-2}$), producing a pronounced decline in SW_{net} (FIG 5a,e). The substantial increase in summer mean LE_{up} , reflecting the higher water consumption of broadleaf forests compared to coniferous forests and grassland, leads to decreases in both H_{up} and LW_{up} , and consequently lowering all three temperature metrics. Results for other temperature variables and regions can be found in Supplementary Tables S1-S5.

TAB 1: Average energy fluxes and temperatures in summer over the Alpine region

Energy fluxes (W/m^2)	$SW_{down,jjaM}$	$SW_{up,jjaM}$	$LW_{up,jjaM}$	$LW_{down,jjaM}$
Ctl	184.49±2.04	25.76±0.32	394.45±1.12	335.26±1.32
Brd	183.15±2.08	28.15±0.36	392.95±1.10	335.35±1.35
Def	186.77±2.07	30.52±0.42	395.31±1.12	335.13±1.35
Aff	181.32±2.13	19.78±0.21	392.71±1.10	335.14±1.33
AfB	179.75±2.04	25.82±0.28	390.02±1.07	335.29±1.34
Energy fluxes (W/m^2)	$LE_{up,jjaM}$	$H_{up,jjaM}$	$G_{down,jjaM}$	
Ctl	55.91±0.68	33.73±0.58	8.38±0.44	
Brd	57.50±0.70	29.51±0.61	8.89±0.48	
Def	52.48±0.61	31.67±0.60	10.33±0.57	
Aff	59.55±0.74	36.31±0.69	6.66±0.29	
AfB	62.36±0.82	28.23±0.66	7.18±0.35	
Temperature (C)	$T_{sfc,jjaX}$	$T_{air,jjaX}$	$T_{atm,jjaX}$	
Ctl	21.71±0.24	18.85±0.22	18.14±0.22	
Brd	21.22±0.23	18.45±0.22	17.78±0.22	
Def	22.37±0.25	18.50±0.22	17.81±0.22	
Aff	20.96±0.23	19.14±0.22	18.39±0.22	
AfB	20.07±0.22	18.47±0.22	17.77±0.21	
Temperature (C)	$T_{sfc,jjaN}$	$T_{air,jjaN}$	$T_{atm,jjaN}$	
Ctl	9.51±0.20	10.75±0.20	11.53±0.21	
Brd	9.44±0.20	10.64±0.20	11.38±0.20	
Def	9.38±0.20	10.91±0.20	11.68±0.21	
Aff	9.41±0.20	10.34±0.20	11.18±0.21	
AfB	9.32±0.20	10.20±0.20	10.97±0.20	
Temperature (C)	$T_{sfc,jjaM}$	$T_{air,jjaM}$	$T_{atm,jjaM}$	
Ctl	15.42±0.21	14.90±0.21	14.91±0.21	
Brd	15.15±0.20	14.64±0.20	14.64±0.20	
Def	15.56±0.21	14.82±0.20	14.83±0.21	
Aff	15.09±0.20	14.83±0.20	14.84±0.21	
AfB	14.61±0.20	14.41±0.20	14.41±0.20	

FIG 5: Biogeophysical impacts of forest change scenarios driven by changes in energy fluxes. Multi-year (2025-2059) summer (June, July, and August) mean daily maximum temperatures (of land surface: T_{sfc} ; of 2-meter air: T_{air} ; and the lowest atmosphere level: T_{atm}), and energy fluxes (down-welling shortwave radiation: SW_{down} ; up-welling shortwave radiation: SW_{up} ; down-welling longwave radiation: LW_{down} ; up-welling longwave radiation: LW_{up} ; latent heat flux from the land to the atmosphere: LE_{up} ; sensible heat flux from the land to the atmosphere: H_{up} ; and ground flux from the land surface to the ground: G_{down}) averaged over the Atlantic region (see Figure 1e) under the present-day forest scenario (Ctl: **a**), and the difference compared to Ctl under the conversion from coniferous to broadleaf forests scenario (Brd: **b**), under the deforestation scenario (Def: **c**), under the forestation scenario (Aff: **d**), and under the combining forestation and conversion from coniferous to broadleaf forests scenario (AfB: **f**). Numbers in blue indicate there is a decrease compared to the corresponding numbers under Ctl, and vice versa for numbers in red. Other temperatures (daily mean and minimum temperatures) and the results of other regions can be found in Table S1-5.

TAB 2: Average energy fluxes and temperatures in summer over the Northern region

Energy fluxes (W/m ²)	SW _{down,jjaM}	SW _{up,jjaM}	LW _{up,jjaM}	LW _{down,jjaM}
Ctl	170.81±2.81	21.15±0.32	397.43±1.49	344.30±1.54
Brd	167.01±2.75	24.69±0.37	394.63±1.45	344.68±1.49
Def	173.03±2.71	27.55±0.41	396.65±1.47	343.62±1.53
Aff	169.05±2.85	17.74±0.27	397.06±1.51	344.49±1.53
AfB	163.52±2.84	23.29±0.37	393.05±1.41	345.02±1.51
Energy fluxes (W/m ²)	LE _{up,jjaM}	H _{up,jjaM}	G _{down,jjaM}	
Ctl	60.26±0.95	27.02±0.66	4.65±0.39	
Brd	62.71±1.09	19.64±0.48	5.54±0.45	
Def	57.72±0.99	22.58±0.58	7.49±0.62	
Aff	61.85±0.93	28.36±0.76	3.96±0.34	
AfB	64.98±1.13	17.98±0.52	4.84±0.41	
Temperature (C)	T _{sfc,jjaX}	T _{air,jjaX}	T _{atm,jjaX}	
Ctl	21.36±0.32	19.63±0.30	19.07±0.30	
Brd	20.41±0.30	18.85±0.29	18.33±0.29	
Def	21.56±0.31	19.01±0.29	18.47±0.29	
Aff	21.15±0.32	19.82±0.31	19.23±0.31	
AfB	19.78±0.29	18.72±0.28	18.19±0.28	
Temperature (C)	T _{sfc,jjaN}	T _{air,jjaN}	T _{atm,jjaN}	
Ctl	10.39±0.25	11.40±0.25	11.97±0.25	
Brd	10.30±0.25	11.25±0.25	11.77±0.25	
Def	10.11±0.25	11.47±0.25	12.07±0.25	
Aff	10.38±0.25	11.22±0.25	11.79±0.26	
AfB	10.27±0.25	11.05±0.25	11.54±0.25	
Temperature (C)	T _{sfc,jjaM}	T _{air,jjaM}	T _{atm,jjaM}	
Ctl	16.00±0.27	15.76±0.27	15.73±0.27	
Brd	15.48±0.26	15.29±0.26	15.25±0.27	
Def	15.87±0.27	15.49±0.26	15.47±0.27	
Aff	15.92±0.27	15.77±0.27	15.73±0.28	
AfB	15.19±0.26	15.12±0.26	15.06±0.26	

TAB 3: Average energy fluxes and temperatures in summer over the Atlantic region

Energy fluxes (W/m^2)	$SW_{down,jjaM}$	$SW_{up,jjaM}$	$LW_{up,jjaM}$	$LW_{down,jjaM}$
Ctl	185.58±2.01	25.82±0.29	408.91±1.50	353.91±1.31
Brd	184.71±1.92	28.40±0.29	407.87±1.49	354.17±1.35
Def	187.32±1.91	29.99±0.30	410.27±1.52	354.26±1.35
Aff	183.30±2.07	21.25±0.24	407.29±1.47	353.45±1.32
AfB	181.47±1.92	26.34±0.26	405.14±1.44	353.97±1.33
Energy fluxes (W/m^2)	$LE_{up,jjaM}$	$H_{up,jjaM}$	$G_{down,jjaM}$	
Ctl	59.87±0.97	38.69±0.69	5.12±0.18	
Brd	62.63±1.14	33.85±0.82	5.04±0.17	
Def	59.22±1.08	35.59±0.75	5.35±0.17	
Aff	60.99±0.95	41.37±0.69	4.79±0.17	
AfB	66.14±1.19	32.12±0.89	4.65±0.16	
Temperature (C)	$T_{sfc,jjaX}$	$T_{air,jjaX}$	$T_{atm,jjaX}$	
Ctl	24.56±0.33	21.67±0.29	20.88±0.29	
Brd	24.14±0.33	21.38±0.29	20.63±0.29	
Def	25.23±0.34	21.46±0.29	20.67±0.29	
Aff	24.03±0.32	21.90±0.29	21.09±0.29	
AfB	23.14±0.31	21.31±0.29	20.57±0.28	
Temperature (C)	$T_{sfc,jjaN}$	$T_{air,jjaN}$	$T_{atm,jjaN}$	
Ctl	12.48±0.22	13.47±0.22	14.01±0.22	
Brd	12.48±0.22	13.45±0.22	13.98±0.22	
Def	12.50±0.22	13.70±0.22	14.25±0.23	
Aff	12.26±0.22	13.05±0.22	13.61±0.22	
AfB	12.29±0.22	13.04±0.22	13.56±0.22	
Temperature (C)	$T_{sfc,jjaM}$	$T_{air,jjaM}$	$T_{atm,jjaM}$	
Ctl	18.06±0.26	17.52±0.25	17.40±0.26	
Brd	17.88±0.26	17.37±0.25	17.26±0.25	
Def	18.29±0.26	17.54±0.25	17.43±0.26	
Aff	17.77±0.26	17.40±0.25	17.28±0.25	
AfB	17.40±0.25	17.11±0.25	17.01±0.25	

TAB 4: Average energy fluxes and temperatures in summer over the Continental region

Energy fluxes (W/m ²)	SW _{down,jjaM}	SW _{up,jjaM}	LW _{up,jjaM}	LW _{down,jjaM}
Ctl	210.45±2.67	29.78±0.36	421.03±1.60	354.92±1.52
Brd	207.83±2.74	31.64±0.40	418.99±1.50	355.02±1.52
Def	211.54±2.71	33.47±0.42	421.94±1.59	354.93±1.53
Aff	208.65±2.84	26.51±0.33	419.17±1.60	354.58±1.52
AfB	204.01±2.70	29.68±0.37	415.83±1.48	354.89±1.51
Energy fluxes (W/m ²)	LE _{up,jjaM}	H _{up,jjaM}	G _{down,jjaM}	
Ctl	70.78±0.80	36.63±1.04	6.81±0.21	
Brd	73.75±0.84	31.44±0.94	6.71±0.21	
Def	69.32±0.79	34.20±0.96	7.21±0.22	
Aff	72.59±0.84	38.27±1.14	6.32±0.20	
AfB	76.97±1.01	29.88±0.97	6.18±0.20	
Temperature (C)	T _{sfc,jjaX}	T _{air,jjaX}	T _{atm,jjaX}	
Ctl	27.48±0.37	24.48±0.31	23.73±0.30	
Brd	26.79±0.34	23.95±0.29	23.24±0.28	
Def	28.02±0.37	24.14±0.30	23.39±0.29	
Aff	26.99±0.37	24.70±0.32	23.92±0.31	
AfB	25.84±0.33	23.81±0.29	23.11±0.29	
Temperature (C)	T _{sfc,jjaN}	T _{air,jjaN}	T _{atm,jjaN}	
Ctl	13.80±0.24	15.03±0.24	15.77±0.25	
Brd	13.70±0.24	14.90±0.23	15.60±0.24	
Def	13.76±0.24	15.26±0.24	16.01±0.24	
Aff	13.51±0.24	14.55±0.24	15.33±0.25	
AfB	13.37±0.24	14.35±0.24	15.08±0.24	
Temperature (C)	T _{sfc,jjaM}	T _{air,jjaM}	T _{atm,jjaM}	
Ctl	20.21±0.27	19.78±0.26	19.82±0.26	
Brd	19.86±0.26	19.46±0.25	19.49±0.25	
Def	20.36±0.27	19.75±0.26	19.77±0.26	
Aff	19.88±0.28	19.59±0.27	19.66±0.27	
AfB	19.31±0.26	19.08±0.25	19.13±0.25	

TAB 5: Average energy fluxes and temperatures in summer over the Southern region

Energy fluxes (W/m ²)	SW _{down,jjaM}	SW _{up,jjaM}	LW _{up,jjaM}	LW _{down,jjaM}
Ctl	268.68±1.65	44.84±0.33	458.60±1.68	361.39±1.55
Brd	268.07±1.70	46.47±0.35	457.49±1.70	361.41±1.54
Def	269.19±1.68	47.93±0.37	459.77±1.70	361.78±1.57
Aff	266.42±1.60	37.71±0.24	456.02±1.67	361.06±1.55
AfB	265.85±1.71	42.93±0.29	453.60±1.71	360.74±1.52
Energy fluxes (W/m ²)	LE _{up,jjaM}	H _{up,jjaM}	G _{down,jjaM}	
Ctl	53.00±1.10	64.76±1.11	6.56±0.16	
Brd	54.50±1.16	62.26±1.16	6.46±0.15	
Def	52.22±1.08	61.95±1.07	6.71±0.17	
Aff	54.06±1.16	71.25±1.24	6.37±0.15	
AfB	56.56±1.42	65.29±1.43	6.17±0.15	
Temperature (C)	T _{sfc,jjaX}	T _{air,jjaX}	T _{atm,jjaX}	
Ctl	37.33±0.37	30.95±0.31	29.80±0.30	
Brd	37.00±0.38	30.67±0.31	29.54±0.30	
Def	37.94±0.38	30.65±0.30	29.49±0.30	
Aff	36.38±0.36	31.45±0.31	30.25±0.30	
AfB	35.66±0.37	30.87±0.32	29.72±0.31	
Temperature (C)	T _{sfc,jjaN}	T _{air,jjaN}	T _{atm,jjaN}	
Ctl	18.09±0.23	19.63±0.23	20.44±0.24	
Brd	18.02±0.23	19.54±0.23	20.33±0.24	
Def	18.08±0.23	19.82±0.23	20.63±0.23	
Aff	17.83±0.23	19.10±0.23	19.98±0.24	
AfB	17.69±0.23	18.93±0.23	19.77±0.24	
Temperature (C)	T _{sfc,jjaM}	T _{air,jjaM}	T _{atm,jjaM}	
Ctl	26.51±0.27	25.09±0.26	25.00±0.26	
Brd	26.33±0.28	24.90±0.26	24.80±0.26	
Def	26.71±0.28	25.06±0.26	24.95±0.26	
Aff	26.06±0.27	24.99±0.26	24.94±0.26	
AfB	25.67±0.28	24.63±0.27	24.57±0.27	

Reviewer 2 Comment 3

What was the rationale for selecting the SSP3-7.0 scenario? Would the conclusions differ under alternative scenarios (e.g., SSP5-8.5)?

Response We thank Reviewer 2 for this question, and therefore added the rationale here. Considering that we used a static land-use map in simulations, the only impact of scenario selection is, indeed, the warming level. We selected SSP3-7.0 because it represents a realistic pathway and a plausible scenario with a pronounced warming signal, allowing us to robustly assess the potential impacts under strong climate change conditions.

Introduction, Line 93-97

For each experiment, we perform a simulation under a Shared Socioeconomic Pathway scenario (SSP3-7.0, as it represents a scenario at the high end of warming that current climate policy could lead to (10)) with a regional climate model (COSMO – CLM²) covering the period from 2015 to 2059.

Reviewer 2 Comment 4

How can cropland be treated? Does it be treated as grassland in various experiments? Or do the analysis focus exclusively on areas covered by natural vegetation?

Response We thank Reviewer 2 for pointing this out, and we apologise for any confusion that was created. We have added some text to the Method section.

Methods, Lines 430-434

Together with the control simulation (Ctl) based on the present-day natural vegetation distribution, we conduct seven simulations in total (Table 1). These experiments only vary in the natural vegetation distribution (coniferous forests, broadleaf forests, and grassland); other land-use types, such as cropland, remain identical in all experiments.

Reviewer 2 Comment 5

L115-118: Why do the Brd and Def simulations exhibit a significant cooling effect on TM (Figure A1) and Tx (Figure A2), yet demonstrate only a limited impact on TN (Figure A3)—particularly during summer—and in some cases, why does the Def test even lead to widespread warming (Figure A3g)? Please provide a detailed explanation of the underlying factors contributing to these differences.

Response We thank Reviewer 2 for highlighting this point, which gives us the chance to clarify the potential reasons behind it. We believe it may be related to diurnal variations in energy fluxes. Unfortunately, the model does not provide sub-daily outputs, so we have added a sentence (see below) to suggest this as a possible explanation. However, due to data limitations, a detailed analysis of this phenomenon is beyond the scope of the present study.

Results, Lines 138-142

Different patterns emerge when considering daily mean and daily minimum temperatures (Figure S3,S4), such as in Southern Europe, both summer mean daily minimum T_{air} and T_{atm} increase under Def (Figure S4g,k). This may be explained by an increase in nighttime H_{up} resulting from reduced LE_{up} under Def, which becomes the dominant factor in the absence of shortwave radiation.

Reviewer 2 Comment 6

Each species occupies a specific ecological niche. If climatic and environmental conditions fail to meet the species' requirements, its survival, reproduction, and expansion may be significantly constrained. As illustrated in Figure 1, the dominant vegetation type in the study area is coniferous forest, with relatively low coverage of broad-leaved forest. Therefore, do the Brd and AfB experiments possess practical guiding significance?

Response We thank Reviewer 2 for pointing it out. Indeed, whether those species can survive under local climate conditions remains a question and an open and much-debated issue. However, in CLM5, within every forest type (being coniferous or broadleaf), there are multiple possible and different tree species. Historical European forest reconstruction datasets also show that broadleaf forests used to dominate over many regions in Europe. Considering that climate conditions may change, we also wrote two sentences to describe this limitation in the revised manuscript version.

Discussion, Lines 314-318

It is evident that different tree species have varying suitable growth regions under specific local climate conditions (11; 12), and climate change could alter their suitability (13). Therefore, for more realistic (e.g. species-level) assessments, future studies could incorporate forest change scenarios that consider the suitability of tree species for future local conditions.

2.2 Minor comments

Reviewer 2 Comment 7

L24: the sentence "e.g.,, the monthly mean daily maximum temperature....." has an extra comma.

Response Thanks for finding this error. We have corrected it.

Reviewer 2 Comment 8

L148: "This warming effect does not exit on TM, JJA and TN, JJA (Figure A7 and Figure A8)" should be modified as "..... (Figure A6 and Figure A8)"?

Response Thanks for pointing out this error. We have rewritten this paragraph.

Reviewer 2 Comment 9

Adjust the vertical axis range of certain graphs (e.g., Figure S23) to facilitate a clearer visualization of variable changes for readers.

Response We thank Reviewer 2 for this suggestion, which has been implemented in Figure S7 and S14.

Reviewer 2 Comment 10

Some variable indices in the figures contain inaccuracies, such as in Figure S17 d-i. Other diagrams also exhibit similar issues. A thorough review is recommended.

Response We thank Reviewer 2 for this issue, and have reworked all figures.

References

- [1] Winckler, J., Reick, C., Luyssaert, S., Cescatti, A., Stoy, P., Lejeune, Q., Raddatz, T., Chlond, A., Heidkamp, M. & Pongratz, J. Different response of surface temperature and air temperature to deforestation in climate models. *Earth System Dynamics Discussions*. **2018** pp. 1-17 (2018)
- [2] Breil, M., Rechid, D., Davin, E., Noblet-Ducoudré, N., Katragkou, E., Cardoso, R., Hoffmann, P., Jach, L., Soares, P., Sofiadis, G. & Others The opposing effects of reforestation and afforestation on the diurnal temperature cycle at the surface and in the lowest atmospheric model level in the European summer. *Journal Of Climate*. **33**, 9159-9179 (2020)
- [3] Breil, M., Weber, A. & Pinto, J. The potential of an increased deciduous forest fraction to mitigate the effects of heat extremes in Europe. *Biogeosciences*. **20**, 2237-2250 (2023)
- [4] Schwaab, J., Davin, E., Bebi, P., Duguay-Tetzlaff, A., Waser, L., Haeni, M. & Meier, R. Increasing the broad-leaved tree fraction in European forests mitigates hot temperature extremes. *Scientific Reports*. **10**, 14153 (2020)
- [5] Pongratz, J., Reick, C., Raddatz, T. & Claussen, M. Biogeophysical versus biogeochemical climate response to historical anthropogenic land cover change. *Geophysical Research Letters*. **37** (2010)
- [6] Devaraju, N., Bala, G. & Nemani, R. Modelling the influence of land-use changes on biophysical and biochemical interactions at regional and global scales. *Plant, Cell Environment*. **38**, 1931-1946 (2015)
- [7] Liu, S., Wang, Y., Zhang, G., Wei, L., Wang, B. & Yu, L. Contrasting influences of biogeophysical and biogeochemical impacts of historical land use on global economic inequality. *Nature Communications*. **13**, 2479 (2022)
- [8] Markonis, Y., Kumar, R., Hanel, M., Rakovec, O., Máca, P. & AghaKouchak, A. The rise of compound warm-season droughts in Europe. *Science Advances*. **7**, eabb9668 (2021)

-
- [9] Zhao, T. & Dai, A. Uncertainties in historical changes and future projections of drought. Part II: model-simulated historical and future drought changes. *Climatic Change*. **144**, 535-548 (2017)
- [10] Hausfather, Z. & Peters, G. Emissions—the ‘business as usual’ story is misleading. *Nature*. **577**, 618-620 (2020)
- [11] De Rigo, D., Caudullo, G., Houston Durrant, T. & San-Miguel-Ayanz, J. The European Atlas of Forest Tree Species: modelling, data and information on forest tree species. *European Atlas Of Forest Tree Species*. pp. 40-45 (2016)
- [12] Casalegno, S., Amatulli, G., Bastrup-Birk, A., Durrant, T. & Pekkarinen, A. Modelling and mapping the suitability of European forest formations at 1-km resolution. *European Journal Of Forest Research*. **130** pp. 971-981 (2011)
- [13] Noce, S., Collalti, A. & Santini, M. Likelihood of changes in forest species suitability, distribution, and diversity under future climate: The case of Southern Europe. *Ecology And Evolution*. **7**, 9358-9375 (2017)